# Analogous mechanism regulating formation of neocortical basal radial glia and cerebellar Bergmann glia

**Xin Heng[1], Qiuxia Guo[1], Alan W Leung[1†], James YH Li[1,2]***

[1]Department of Genetics and Genome Sciences, University of Connecticut School of Medicine, Farmington, United States; [2]Institute for Systems Genomics, University of Connecticut, Farmington, United States

**Abstract** Neocortical basal radial glia (bRG) and cerebellar Bergmann glia (BG) are basal progenitors derived from ventricular apical radial glia (aRG) that selectively lose their apical processes. bRG and BG have been implicated in the expansion and folding of the cerebrum and cerebellum, respectively. Here, we analyzed the molecular characteristics and development of bRG and BG. Transcriptomic comparison revealed striking similarity of the molecular features of bRG and BG. We found that heightened ERK signaling activity in aRG is tightly linked to the temporal formation and the relative abundance of bRG in human and mouse cortices. Forced activation of an FGF-ERK-ETV axis that is crucial to BG induction specifically induced bRG with canonical human bRG features in mice. Therefore, our data point to a common mechanism of bRG and BG generation, bearing implications to the role for these basal progenitors in the evolution of cortical folding of the cerebrum and cerebellum.

*For correspondence: jali@uchc.edu

Present address: †Department of Genetics, Yale University, New Haven, United States

Competing interests: The authors declare that no competing interests exist.

## Introduction

The evolutionary expansion and elaboration of a gyrated neocortex underlie the complex cognition and development of intellect that characterize primates, especially humans. Emerging evidence suggests that differences in the type, abundance, and mode of division of neural stem and progenitor cells contribute to the diversity in the size and shape of the mammalian neocortex across species (*Lui et al., 2011*; *Paridaen and Huttner, 2014*; *Sun and Hevner, 2014*; *Dehay et al., 2015*). At the onset of neurogenesis, neuroepithelial cells (neural stem cells) become radial glia. These more committed neural progenitor cells reside in the ventricular zone (VZ) and are referred to as apical radial glia (aRG). Their apical processes are integrated into the ventricular surface while their basal fibers extend radially to the pial basement membrane (*Götz and Huttner, 2005*). aRG generate neurons directly or indirectly via becoming intermediate progenitor cells (IPC) that occupy the subventricular zone (SVZ). In contrast to their limited proliferation potential in lissencephalic rodents (*Fietz and Huttner, 2011*), IPC undergo multiple rounds of self-renewing division in species with enlarged and folded neocortices (*Fietz et al., 2010*; *Hansen et al., 2010*; *Betizeau et al., 2013*; *Gertz et al., 2014*). The increased proliferation of IPC expands the SVZ, which is subdivided into inner and outer compartments in primates (*Smart et al., 2002*; *Zecevic et al., 2005*). The outer SVZ may be responsible for the increased number of upper-layer neurons and thus the tangential expansion of neocortical surface area (*Lui et al., 2011*; *Reillo et al., 2011*; *Reillo and Borrell, 2012*).

Recent advances in cell labeling and imaging have identified other basal progenitors in addition to IPC. At mid-neurogenesis, some aRG selectively lose their apical processes and move their soma to the outer SVZ (*Fietz et al., 2010*; *Hansen et al., 2010*; *Reillo et al., 2011*; *Martínez-*

**eLife digest** The outer layer of the brain of a mammal, called the cortex, helps support mental abilities such as memory, attention and thought. In rodents, the cortex is smooth whereas in primates it is organized into folds. These folds increase the surface area of the brain and thus the number of neurons it can contain, which may in turn increase its processing power. Folding occurs as the brain develops in the womb. Specialized cells called basal or outer radial glia, which are more abundant in humans than in rodents, are believed to trigger the folding process.

Another area of the brain, called the cerebellum, is intricately folded in both rodents and humans. As the brain develops, cells within the cerebellum called Bergmann glia cause the tissue to fold. Bergmann glia and basal radial glia share a number of similarities, but it was not known whether the same molecular pathway might regulate both types of cell.

Now, Heng et al. show that Bergmann glia in the cerebellums of mice and basal radial glia in human cortex contain similar sets of active genes. Moreover, the molecular pathway that gives rise to Bergmann glia in mice is also active in the cortex of both mice and humans. However, it is much more active in humans, leading Heng et al. to speculate that high levels of activity in this pathway might give rise to basal radial glia. Consistent with this prediction, artificially activating the pathway at high levels in mouse cortex triggered the formation of basal radial glia in mice too. These results thus suggest that a common mechanism generates both types of glial cells involved in brain folding.

The work of Heng et al. lays the foundations for further studies into how these cells fold the brain and thus how they contribute to more complex mental abilities. Remaining questions to address are whether other species with Bergmann glia also have folded cerebellums, and whether incorrect development of basal radial glia in humans leads to disorders in which the cortex folds abnormally.

*Martínez et al., 2016*). These newly generated basal radial glia (bRG, also called outer radial glia) are abundantly present in gyrencephalic cortex (*Fietz et al., 2010*; *Hansen et al., 2010*; *Reillo et al., 2011*) but are relatively rare in the lissencephalic mouse cortex (*Shitamukai et al., 2011*; *Wang et al., 2011*). It has thus been speculated that bRG expansion is responsible for the emergence and evolution of cortical convolutions (*Fietz and Huttner, 2011*; *Lui et al., 2011*; *Reillo et al., 2011*; *Reillo and Borrell, 2012*; *Geschwind and Rakic, 2013*; *Nonaka-Kinoshita et al., 2013*; *Borrell and Götz, 2014*; *Dehay et al., 2015*; *Fernández et al., 2016*). However, the direct link of bRG to cortical gyrification remains unclear because bRG abundance does not correlate with either gyrification or the phylogeny of the neocortex (*García-Moreno et al., 2012*; *Hevner and Haydar, 2012*; *Kelava et al., 2012*). In mice, genetic manipulations of a number of intrinsic factors (*Stahl et al., 2013*; *Florio et al., 2015*; *Wong et al., 2015*; *Ju et al., 2016b*) or signaling pathways (*Lui et al., 2014*; *Pollen et al., 2015*; *Wang et al., 2016*) enhance the generation of bRG as well as IPC. In some of these experiments, expansion of these basal progenitors resulted in the folding of the mouse neocortex (*Stahl et al., 2013*; *Florio et al., 2015*; *Wong et al., 2015*; *Wang et al., 2016*), highlighting the importance of bRG and IPC in cortical convolution. Yet little is known about the molecular events that specifically control the transition of aRG into bRG. This information is crucial to our understanding of the molecular basis for the greater abundance of bRG in gyrencephalic than in lissencephalic cortices. It is also important to determine the precise contribution of bRG, relative to IPC, to cortical gyrification.

In amniotes, the folding of the cerebellar cortex results in the formation of an elaborate set of folia similar to neocortical gyri. More extensive folding of the cerebellar cortex correlates with more complex behaviors (*Iwaniuk et al., 2006*; *Lisney et al., 2008*; *Hall et al., 2013*). From sharks to primates, the cerebellum and neocortex grow regularly and disproportionately to the rest of the brain, with the extent of gyrification reflecting the size of these structures (*Yopak et al., 2010*). These observations suggest that folding of the cerebral and cerebellar cortex is an evolutionary adaptation that allowed the enlargement of brain surface area and thereby the accommodation of more complex functions. Expansion of the granule cell precursors that reside in the external granule layer (EGL) is thought to be the primary driver of cerebellar foliation (*Corrales et al., 2004*, *2006*; *Sudarov and Joyner, 2007*). However, emerging evidence suggests that the interaction

between BG and basement membrane is important for cerebellar foliation (*Belvindrah et al., 2006*; *Mills et al., 2006*; *Qiu et al., 2010*; *Ma et al., 2012*). We recently discovered that the deletion of *Ptpn11*, which codes for the protein tyrosine phosphatase Shp2, blocks BG formation and cerebellar foliation, whereas EGL proliferation remains relatively normal in perinatal mouse cerebella (*Li et al., 2014*). The expression of a constitutively active Mek1 (Map2k1), Mek1$^{DD}$ (*Cowley et al., 1994*), which specifically activates ERK, rescues both cerebellar foliation and BG generation (*Li et al., 2011*), demonstrating the critical role of ERK signaling in BG induction and the essential role of BG in cerebellar foliation. Similar to the cytogenesis of bRG, nascent BG are derived from cerebellar aRG between embryonic day (E) 13.5 and E17.5, again by selectively losing their apical processes and relocating their soma to a basal position (but in this case to the prospective Purkinje cell layer) (*Yuasa, 1996*; *Yamada and Watanabe, 2002*). These nascent BG continue to proliferate until postnatal day (P) seven when they exit the cell cycle and become mature BG (*Parmigiani et al., 2015*). After their induction, BG express neural stem cell markers, such as Sox2, Sox9, and Tnc through adulthood (*Sottile et al., 2006*; *Alcock and Sottile, 2009*; *Koirala and Corfas, 2010*). The increase and rearrangement of BG basal fibers are associated with dramatic expansion of the cerebellar cortex and fissure formation in perinatal stages (*Sudarov and Joyner, 2007*). Several bRG-signature genes, such as *TNC*, *FABP7*, and *PTPRZ1* (*Pollen et al., 2015*; *Thomsen et al., 2016*), are also well-known BG markers (*Feng et al., 1994*; *Yuasa, 1996*; *Tanaka et al., 2003*). These observations raise the question whether the formation of bRG and BG is controlled by related mechanisms involving ERK signaling.

In the current study, we investigated the hypothesis that conserved mechanisms regulate the transition of aRG to BG in the cerebellum, and to bRG in the neocortex. We first established the transcriptomic profile of nascent BG in the mouse cerebellum, and then compared the molecular features of BG to those of human bRG that were recently identified by single-cell RNA sequencing (seq) (*Pollen et al., 2015*; *Thomsen et al., 2016*). Given the crucial role of ERK signaling in the aRG-to-BG transition in the mouse cerebellum (*Li et al., 2014*), we analyzed and compared the ERK signaling activity in cortical aRG during human and mouse corticogenesis. Finally, we tested whether activation of ERK induced bRG in the mouse neocortex. Our findings demonstrate that BG and bRG not only share similar molecular features, but also related molecular mechanisms in their generation.

## Results

### bRG-specific genes are enriched in BG

Using single-cell RNA-seq, two groups have independently identified specific molecular markers of bRG in human fetal neocortex (*Pollen et al., 2015*; *Thomsen et al., 2016*). Using a new computational pipeline (*Guo et al., 2015*), we identified the consensus signatures for bRG based on the published single-cell RNA-seq datasets (*Figure 1A*). Our analysis confirmed most of the reported bRG markers (*Pollen et al., 2015*; *Thomsen et al., 2016*) and identified additional ones (*Supplementary file 1A*). Inspection of in situ hybridization data of the Allen Mouse Brain Atlas revealed that more than half (51.5%, n = 66) of these bRG-signature gene orthologs were specifically or highly expressed in BG of mouse cerebellum at P56 (*Figure 1B*, and *Supplementary file 1B*). By contrast, only 2.2% and 4.5% of two randomly selected gene groups were detected in BG (n = 44 and 46, Fisher's exact test, p<0.001, see Method section for details of unbiased expression analysis). Inspection of the Allen Mouse Developmental Brain revealed that many bRG markers appeared to be expressed in BG of the mouse cerebellum at P4 when immature BG were relatively easy to be identified (*Figure 1—figure supplement 1A*). In agreement with RNA in situ hybridization data, immunofluorescence revealed that Hopx, a specific bRG marker (*Pollen et al., 2015*; *Thomsen et al., 2016*), was co-localized with the BG marker Fabp7 (also known as BLBP) and Sox2, in the mouse cerebellum at E14.5 and E18.5 (*Figure 1C*). As Hopx, Fabp7 and Sox2, like many other BG markers, are also expressed in the VZ, we define BG as those triple labeled cells that have delaminated from the VZ at E14.5 or those reside in the Purkinje cell layer with radial fibers in the molecular layer. Moreover, antibody staining for Hopx and Fabp7 and in situ hybridization for *Etv4*, *Etv5*, *Ptprz1*, *Tnc1*, and *Slc1a3* showed that these bRG markers were absent from the cerebellar cortex of *Ptpn11*-deficient cerebella, where BG generation is blocked (*Li et al., 2014*), at E16.5

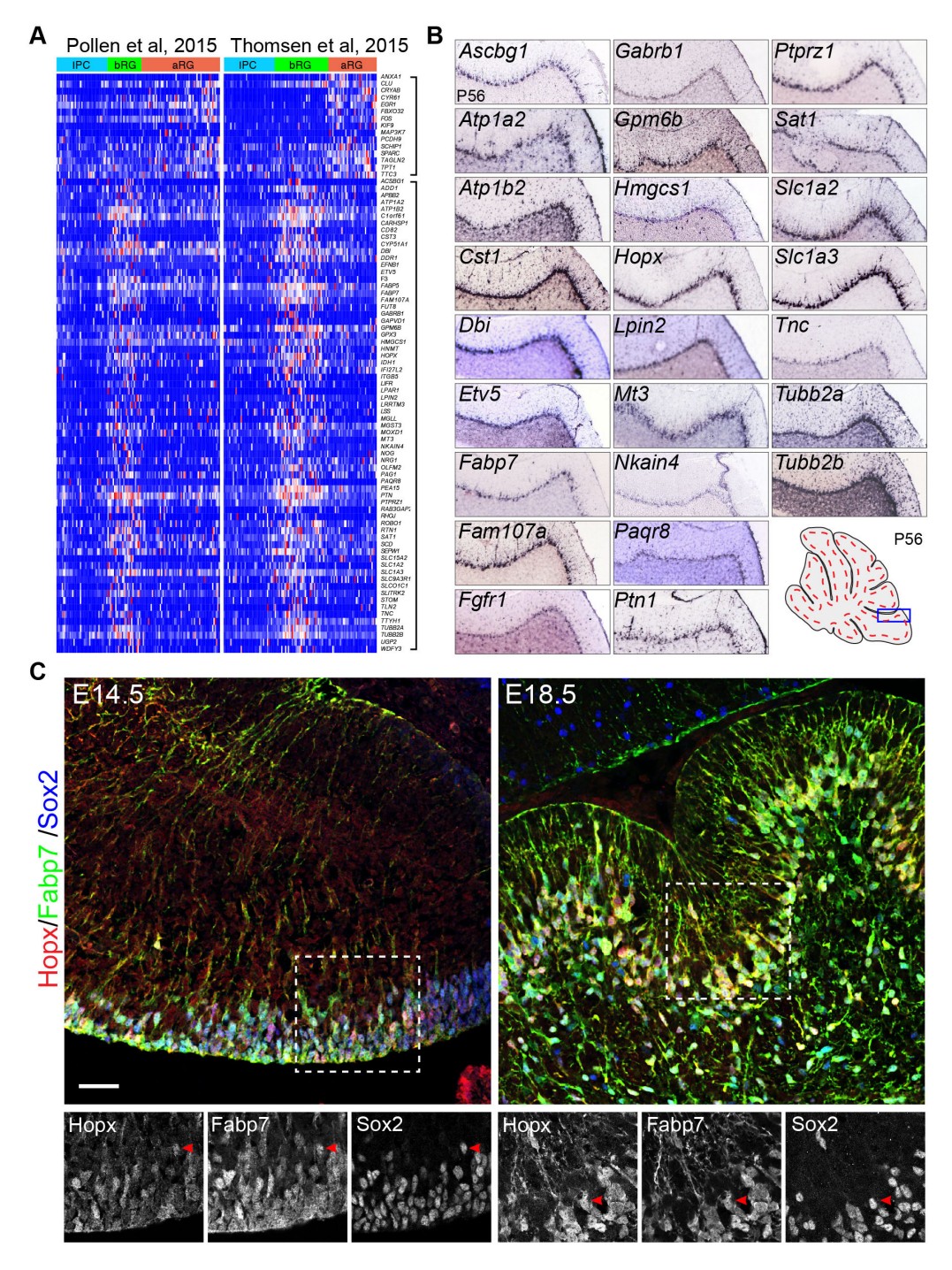

**Figure 1.** Mouse orthologs of human basal radial glia (bRG)-signature genes are specifically or highly expressed in Bergmann glia (BG) of mouse cerebella. (A) Heatmap showing the consensus signature gene sets for apical radial glia (aRG, top bracket to the left) and bRG (lower bracket to the left) in the single-cell RNA-seq datasets of Pollen *et al.* and Thomsen *et al.* Genes of the aRG and bRG gene sets are mostly not expressed in intermediate progenitor cells (IPC). (B) In situ hybridization of selected bRG-specific genes in BG of P56 mouse cerebella. The images were generated by the Allen Institute for Brain Science (*Lein et al., 2007*). (C) Immunofluorescent staining of Hopx, Fabp7, and Sox2 shows that Hopx is expressed in BG in E14.5 and E18.5 mouse cerebella. The boxed areas are enlarged and shown in separate channels below; arrowheads point to BG triple-labeled with Hopx, Fabp7, and Sox2. Scale bar: 40 µm.

The following figure supplements are available for figure 1:

*Figure 1 continued on next page*

*Figure 1 continued*

**Figure supplement 1.** bRG-specific genes are expressed in wild-type but not *Ptpn11-cKO* cerebella.

**Figure supplement 2.** Inspection of gene expression in P56 mouse cerebella.

(*Figure 1—figure supplement 1B and C*). Collectively, our data suggest that bRG and BG have similar molecular features.

We next sought to investigate whether nascent BG also have similar molecular features as the human fetal bRG through genome-wide expression analysis. Previous studies have demonstrated that nascent BG are initially formed at E13.5 (*Yuasa, 1996*), and their generation is blocked by *Ptpn11* deletion but rescued by *Mek1$^{DD}$* expression (*Li et al., 2011*). Therefore, we reasoned that the level of BG-enriched transcripts would increase from E12.5 to E14.5 in wild-type (WT) cerebella, decrease in *Ptpn11* conditional knock-out (cKO) cerebella, and become normal in *Ptpn11*-cKO cerebella with *Mek1$^{DD}$* expression (*Ptpn11-cKO;Map2k1*). Through RNA-seq, differentially expressed genes were identified by pairwise comparisons among the cerebella of these three genotypes at E12.5, E13.5, and E14.5 (*Supplementary file 2*). Validation with qPCR and in situ hybridization confirmed the overall accuracy of the RNA-seq data (*Figure 2—figure supplement 1*). As expected, markers for BG, but not other major cerebellar cell types were significantly decreased in *Ptpn11*-cKO cerebella compared to the control at E13.5 and E14.5 (*Figure 2—figure supplement 2A and B*). By using an intersection-of-list approach based on the above-mentioned logic, we identified 117 putative BG-specific genes (*Figure 2A*), 35.5% of which were apparently expressed in BG in P56 mouse cerebella (Fisher's exact test to compare random gene sets, p<0.001; *Supplementary file 1C*) according to Allen Brain Atlas (*Lein et al., 2007*).

Because the intersection-of-list approach is difficult to control for a type I error (*Natarajan et al., 2012*), we conducted an unsupervised weighted gene coexpression network analysis (WGCNA). WGCNA can elucidate the higher-order relationships between genes based on their coexpression relationships and thus allows the delineation of modules – groups of genes with highly correlated expression patterns, which represent biologically related genes or cell-type-specific markers (*Oldham et al., 2008*; *Lui et al., 2014*). A coexpression module that was enriched for the 117 BG-specific genes was identified (*Figure 2B*). In agreement with our prediction, the BG-module eigengene values, which summarize the overall expression profile of genes in a module (the first principal component) (*Langfelder and Horvath, 2008*), gradually increased from E12.5 to E14.5, was significantly lower in *Ptpn11*-cKO cerebella than the control at each stage, and became comparable between *Ptpn11*-cKO; *Map2k1* and WT cerebella (*Figure 2C*). BG-module hub genes (those most highly connected within the module) were enriched for functions in stem cell development and Map2k1-ERK signaling (*Figure 2D*), in agreement with the critical role of this signaling pathway in the generation of BG as presumptive neural progenitors in the cerebellum. 98 of the 117 BG-specific genes identified by the intersection-of-list approach were included in the 334 BG-module hub genes, suggesting that WGCNA discovers additional BG-enriched genes that are missed by the intersection-of-list method. Indeed, we identified additional 29 genes that are apparently expressed in BG at P56 according to the Allen Mouse Brain (*Supplementary file 1D*).

We then compare the high-confidence BG-specific genes with bRG-signature markers with three statistical methods. Hypergeometric distribution revealed significantly overlap between BG module hub genes and bRG markers (p=1.39 $\times$ 10$^{-7}$). Moreover, gene set enrichment analysis (*Subramanian et al., 2005*) showed that bRG marker genes were significantly down-regulated in *Ptpn11*-cKO cerebella at E13.5 (*Figure 2—figure supplement 1C and D*). Finally, a pathway and gene set over dispersion analysis (PAGODA) was performed on the single-cell RNA-seq datasets (*Pollen et al., 2015*; *Thomsen et al., 2016*) using extensive gene sets, including the newly identified putative BG and bRG markers, to identify cells that exhibit a statistically significant excess of coordinated variability (*Fan et al., 2016*). Notably, both the consensus bRG signature gene (*Figure 1A*) and BG-specific gene sets identified the same population of human cortical progenitor cells (presumably bRG, *Figure 2E*). Collectively, these data show that the gene expression signatures of neocortical bRG and nascent murine BG are remarkably similar.

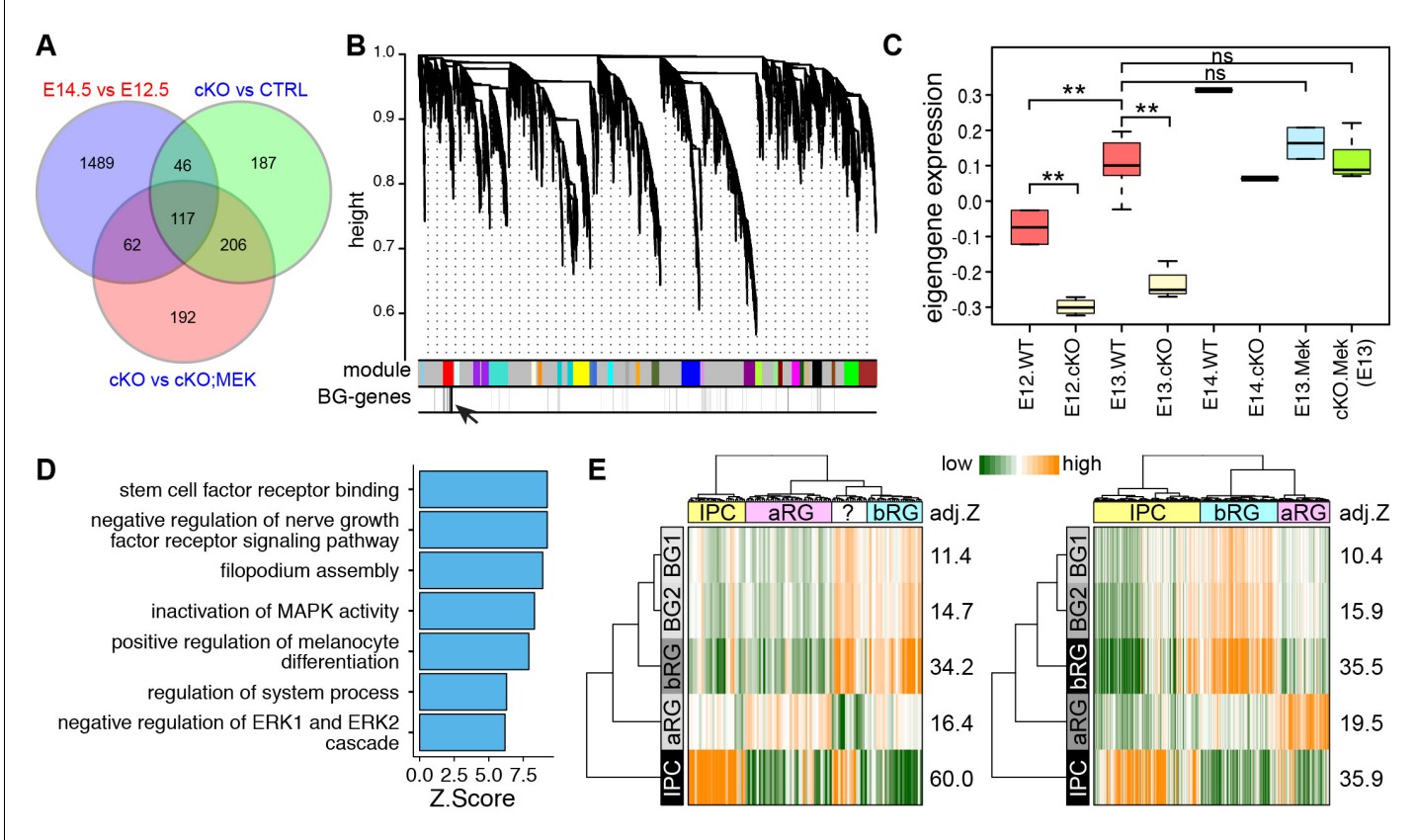

**Figure 2.** The molecular features of newly generated BG are similar to those of human bRG. (**A**) Identification of BG candidate genes by intersecting significantly up- (red) and down- (blue) regulated genes between different embryonic stages and/or genotypes as indicated. (**B**) Dendrograms showing average linkage hierarchical clustering of genes on the basis of topological overlap. Modules of coexpressed genes are assigned in color blocks, as indicated by the horizontal bar beneath the dendrograms; the second bar shows BG-specific genes, indicated by individual vertical lines; the arrow indicates the aggregation of vertical lines – enrichment of BG-specific genes – corresponding to the BG module. (**C**) Boxplots showing module eigengene expression of the BG-module in different genotypes and embryonic stages. **$p<0.01$ (ANOVA with a post-hoc Turkey-Kramer multiple comparison test); ns, not significant ($p>0.05$). (**D**) Functional enrichment of BG-module hub genes. (**E**) PAGODA analyses of Pollen's (left) and Thomsen's (right) single-cell RNA-seq datasets show that human cortical bRG are significantly enriched for the BG-module genes detected by PAGODA. The question mark indicates an undefined cell type. Adjusted $Z$-scores (a $Z$-score $>1.96$ is equivalent to $p<0.05$) are shown to the left of the heatmap.

The following figure supplements are available for figure 2:

**Figure supplement 1.** Validation of RNA-seq data.

**Figure supplement 2.** The loss of *Ptpn11* greatly reduces the expression of BG, but not non-BG, marker genes.

## Stronger FGF-ERK signaling activity in human than mouse neocortical aRG

We also identified 15 human cortical aRG-specific markers based on the published single-cell RNA-seq datasets (***Figure 1A*** and ***Supplementary file 1A***). Among them, *FOS* and *EGR1*, two early-response targets of FGF-ERK signaling (***Kang et al., 2005***), are expressed in human but not mouse aRG (***Pollen et al., 2014***). To identify ERK early-response gene, we mined published microarray data profiling the transcriptional responses to ERK activation in mouse embryonic stem cells (***Hamilton and Brickman, 2014***). We identified genes that were significantly up-regulated (at least 2-fold; adjusted $p<0.05$) within 4 hr after ERK activation (***Supplementary file 3***). Out of the 15 aRG-specific markers, seven genes, *EGR1*, *FOS*, *ANXA1*, *CLU*, *CRYAB*, *CYR61*, and *FBXO32*, were among the early-response genes induced by ERK signaling (***Figure 3A***). Remarkably, orthologs of

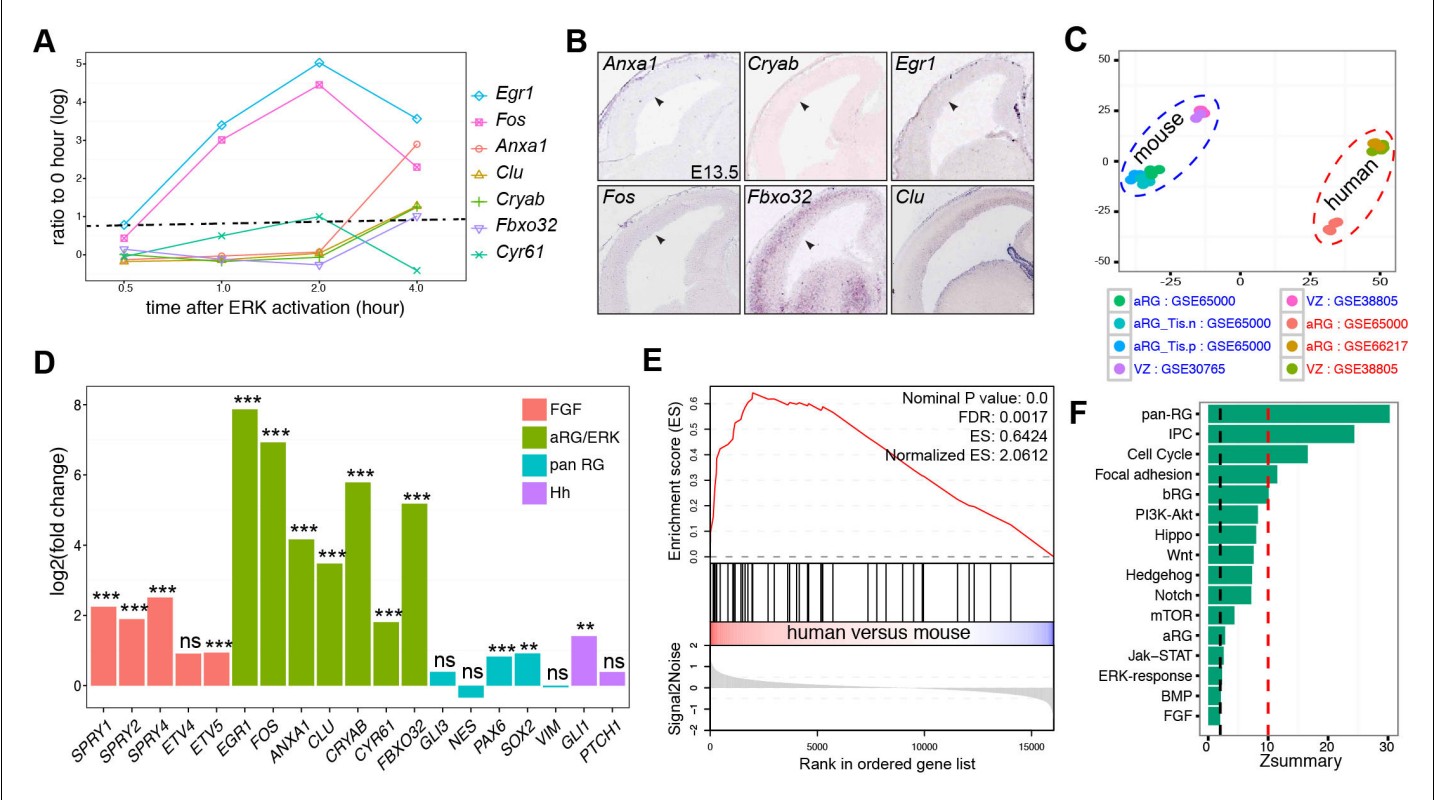

**Figure 3.** Heightened ERK signaling activity in human aRG. (**A**) Line-graph showing the fold increases (log$_2$ scale) in ERK-responding genes between time zero and the indicated time points after ERK activation, based on a microarray time course analysis (**Hamilton and Brickman, 2014**). The fold changes above the dashed line are statistically significant (adjusted p<0.05). (**B**) In situ hybridization of mouse cortical sections at E14.5. Images were obtained from GenePaint (**Visel et al., 2004**). (**C**) Principal component plot showing the relationship among samples: GSE30765 (**Ayoub et al., 2011**), GSE38805 (**Fietz et al., 2012**), GSE65000 (**Florio et al., 2015**), and GSE66217 (**Johnson et al., 2015**). (**D**) RNA-seq analysis of FGF-ERK read-out genes in aRG from human and mouse neocortex. **p<0.005; ***p<0.001; ns, not significant. (**E**) Gene Set Enrichment Analysis plot shows the significant enrichment of early-response (within 2 hr) genes induced by ERK activation in human versus mouse aRG. (**F**) Preservation of pathways and gene sets in human and mouse cortical coexpression networks. The blue and red dashed lines indicate Zsummary at 2 and 10, which are the cutoff for not significant and highly significant, respectively.

The following figure supplement is available for figure 3:

**Figure supplement 1.** Comparison of gene expression in aRG/VZ between human and mouse neocortex.

these human aRG-specific genes, with the exception of *Cy61* and *Clu*, were transcriptionally silent in the VZ of mouse neocortex at E14.5 (**Figure 3B**). These data suggest the presence of robust ERK signaling activity in human, but not mouse, aRG during mid-neurogenesis of the developing cortex.

To systematically compare FGF-ERK signaling activities in the neocortical aRG between humans and mice, we performed differential expression analyses based on six published RNA-seq datasets generated from laser-capture microdissected VZ (**Ayoub et al., 2011**; **Fietz et al., 2012**) or sorted aRG (**Florio et al., 2015**; **Johnson et al., 2015**). Among the 21 E14.5 mouse and 13 GW13–18 human aRG samples included in these independent studies, there was a close similarity among samples of the same species (**Figure 3C** and **Figure 3—figure supplement 1A**). On the other hand, there was a significant positive correlation (Spearman's rank correlation coefficient $R = 0.85$, $p = 1 \times 10^{-200}$) between the mean expression levels in mouse and human aRG (**Figure 3—figure supplement 1B**), demonstrating the compatibility of the mouse and human datasets. The percentages of up and down regulated genes in the human and mouse samples were similar (**Figure 3—figure supplement 1C**). Across a panel (n = 56) of pan-aRG markers (**Lui et al., 2014**), 37.9% were

up-regulated (17.3% down) in human aRG whereas 44.8% were unchanged (*Figure 3D* and *Figure 3—figure supplement 1E*). Collectively, these results demonstrated the absence of systematic bias between the two species in gene expression measurements by RNA-seq. The mRNA levels of FGF-readout genes (*SPRY1, SPRY2, SPRY4,* and *ETV5*) and of all seven ERK early-response genes were significantly higher in humans than those in mice (*Figure 3D*). In agreement with a recent report (*Wang et al., 2016*), the hedgehog readout gene *Gli1* (but not *Ptch1*) was moderately increased in human versus mouse aRG (*Figure 3D*). To analyze pathways differentially enriched in either human or mouse aRG, we performed a gene set enrichment analysis (GSEA) (*Subramanian et al., 2005*). In agreement with earlier findings (*Fietz et al., 2012*; *Florio et al., 2015*), GSEA showed that the gene sets significantly enriched in human aRG contain genes encoding hallmarks of epithelial-mesenchymal transition, extracellular matrix components, extracellular matrix receptors, and integrin signaling, whereas those enriched in mouse aRG are involved in transcription, translation, DNA replication, and the citric acid cycle (*Supplementary file 4A*). Among the most highly enriched genes in human aRG were early-response genes induced by ERK (NES = 2.06, FDR = 0.001) and BG-module genes (NES = 1.91, FDR = 0.01) (*Figure 3E* and *Supplementary file 4A*). These findings indicate elevated FGF-ERK signaling activity in human cortical aRG compared to the mouse cortical aRG.

To further study the difference between mouse and human corticogenesis, we used a complementary approach in which the higher-order relationships between genes were compared based on their coexpression relationships in separate human and mouse networks. Changes in network position between species can reveal divergent regulation or a novel function that contributes to human-specific phenotypes (*Oldham et al., 2006*; *Miller et al., 2010*; *Lui et al., 2014*). We used WGCNA, which had been extensively applied to compare corticogenesis across species (*Oldham et al., 2006*; *Miller et al., 2010*; *Konopka et al., 2012*; *Lui et al., 2014*) and under pathological conditions (*Horvath et al., 2006*; *Chen et al., 2008*; *Oldham et al., 2012*; *Shirasaki et al., 2012*), to study the preservation of signaling pathways (KEGG and Reactome), pan-RG signature genes (*Lui et al., 2014*), early-response genes induced by ERK, and signature genes for aRG, bRG, and IPC (*Figure 1A*). As expected, the strongly preserved gene sets (Zsummary sore >10) were those related to pan-aRG, IPC, and cell cycle (*Figure 3F*). Of note, the gene sets related to aRG, ERK-early-response genes, FGF and BMP signaling pathway were weakly preserved between the human and mouse coexpression networks (*Figure 3F*). This suggests that the transcriptional regulation of aRG and FGF-ERK pathway genes greatly differs between human and mouse corticogenesis.

## Formation of bRG correlates with intensified ERK signaling activity

To corroborate our bioinformatic analysis, we performed immunohistochemistry of human and mouse embryonic cortices using an anti-phospho-ERK (pERK) antibody that detects both phosphorylated ERK1 and ERK2. We detected robust pERK immunoreactivity in the soma and radial fibers of radial glia marked by Pax6 in the VZ and outer SVZ of human fetal brain gestation week (GW) 19, when bRG are present in abundance (*Hansen et al., 2010*; *Pollen et al., 2015*) (*Figure 4A and B*). Furthermore, abundant pERK/HOPX double-labeled cells were found in the outer SVZ (*Figure 4A and C*). In mouse embryonic brains, in contrast to the strong pERK immunoreactivity seen in the VZ of the ventral telencephalon, ventral mid-hindbrain, and cerebellum, only weak and diffuse pERK signals were detected in the dorsal telencephalon between E13.5 and E16.5 (*Figure 4D and F*). The cortical pERK expression pattern was similar to that described in the previous publications (*Faedo et al., 2010*; *Pucilowska et al., 2012*). Remarkably, by E17.5 strong pERK signal was detected in aRG and some basally located cells in the neocortex (*Figure 4E and F*). Although weak Hopx expression was detected in the VZ in the E16.5 cortex, few Hopx-positive (Hopx[+]) cells were found in the SVZ until E17.5 (*Figure 4G,H*, and data not shown). Between E17.5 and E18.5, scattered Hopx[+] cells were present in the SVZ and cortical plate; the colocalization for Fabp7 and Sox2 (*Figure 4G*) suggested that these cells were mouse bRG. Notably, pERK and Hopx colocalized in basally located cells, with about half of them displaying unipolar processes that were extending to the pial basement membrane (unipolar, 50.21 ± 4.76%; bipolar, 7.55 ± 9.67%; the rest were ambiguous or multipolar; n = 1,246; *Figure 4H and I*). Our data suggest that the development of presumptive bRG correlates closely with ERK

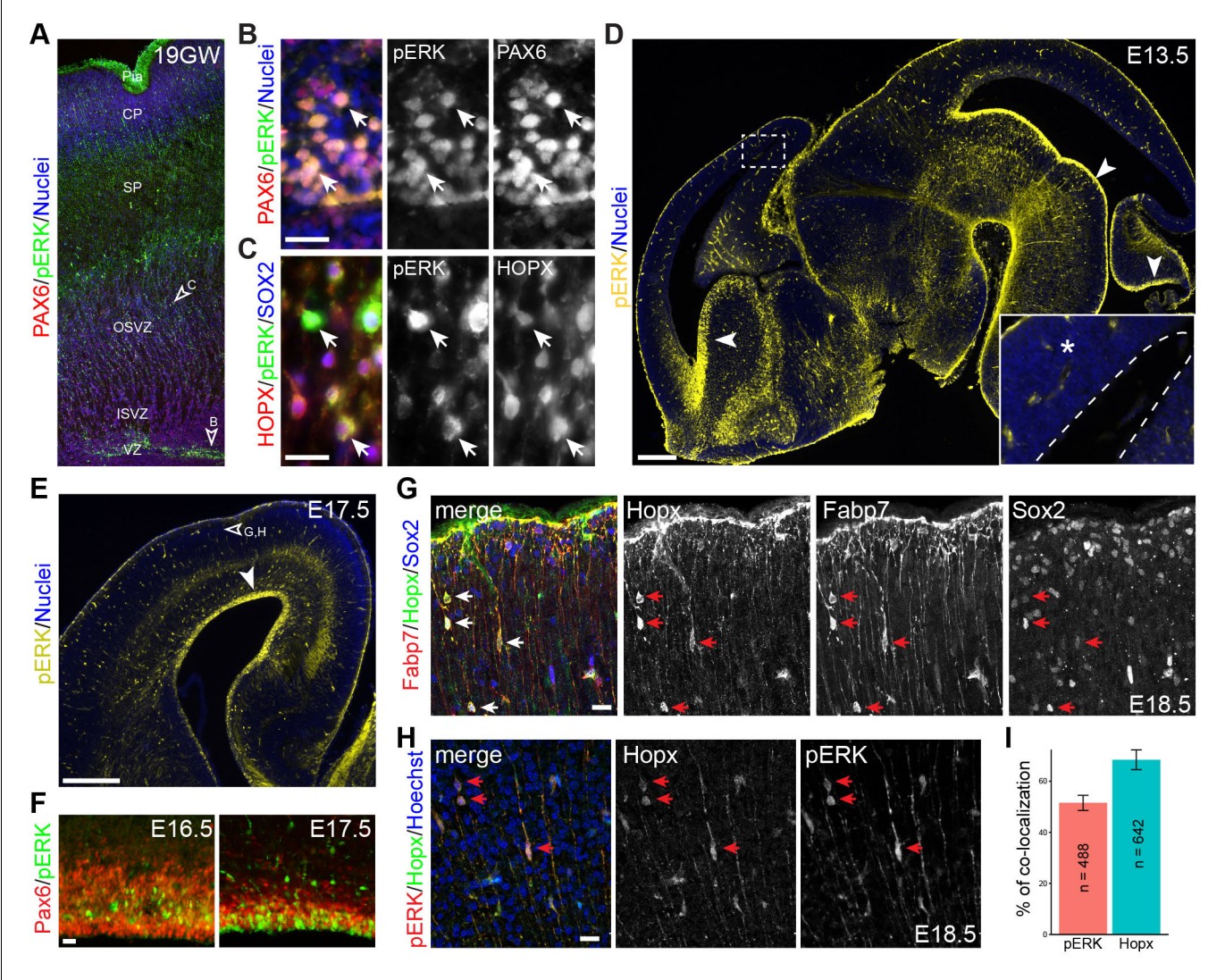

**Figure 4.** Activation of ERK signaling is associated with bRG formation. (A–H) Immunofluorescence on sections of human (A–C, 19 gestational week) and mouse (D–H) fetal brains. Arrowheads indicate the regions that are enlarged in B and C and show pERK immunoreactivity in Pax6-positive aRG (B) and Hopx/Sox2 doubled-labeled bRG (C); the boxed areas in D are enlarged in the inset; the dashed line demarcates the ventricle; arrowheads point to robust pERK staining in the ventricular zone (VZ); the unfilled arrowhead in E indicates the area enlarged in G and H. Note that pERK signals are increased at the apical surface of VZ from E16.5 to E17.5 (F). Arrows indicate Hopx/Fabp7/Sox2 triple-positive cells (G) and Hopx/pERK double-labeled cells (H) on E18.5 cortical sections. Nuclei are stained with Hoechst 33342. (I) Bar charts showing the percentage of Hopx/pERK double-labeled cells relative to the total number of counted pERK (n = 642) or Hopx (n = 488) cells. Abbreviations: cp, cortical plate; ISVZ, inner subventricular zone; OSVZ, outer subventricular zone; Pia, pial surface; SP, cortical subplate. Scale bars: 20 μm (B, C, G, H and F), 200 μm (D), and 100 μm (E).

signaling in both human and mouse cortex, albeit the delayed appearance and reduced number of bRG in the mouse neocortex.

## Hyperactivation of ERK signaling induces bRG in the murine neocortex

The function of the ERK pathway in bRG formation was tested by expressing the ERK activator $Mek1^{DD}$ in the E14.5 mouse cortex through *in utero* electroporation (*Shimogori and Ogawa, 2008*). The electroporation of GFP had little effect on the expression of bRG markers; however, bRG markers including Hopx, Fabp7, *Tnc*, *Slc1a3*, and *Ptprz1* were robustly induced following the electroporation of $Mek1^{DD}$ 48 hr after electroporation (*Figure 5A–C*, *Figure 5—figure supplement 1*, and *Video 1*). As reported previously (*Maiorano and Mallamaci, 2009*), electroporation

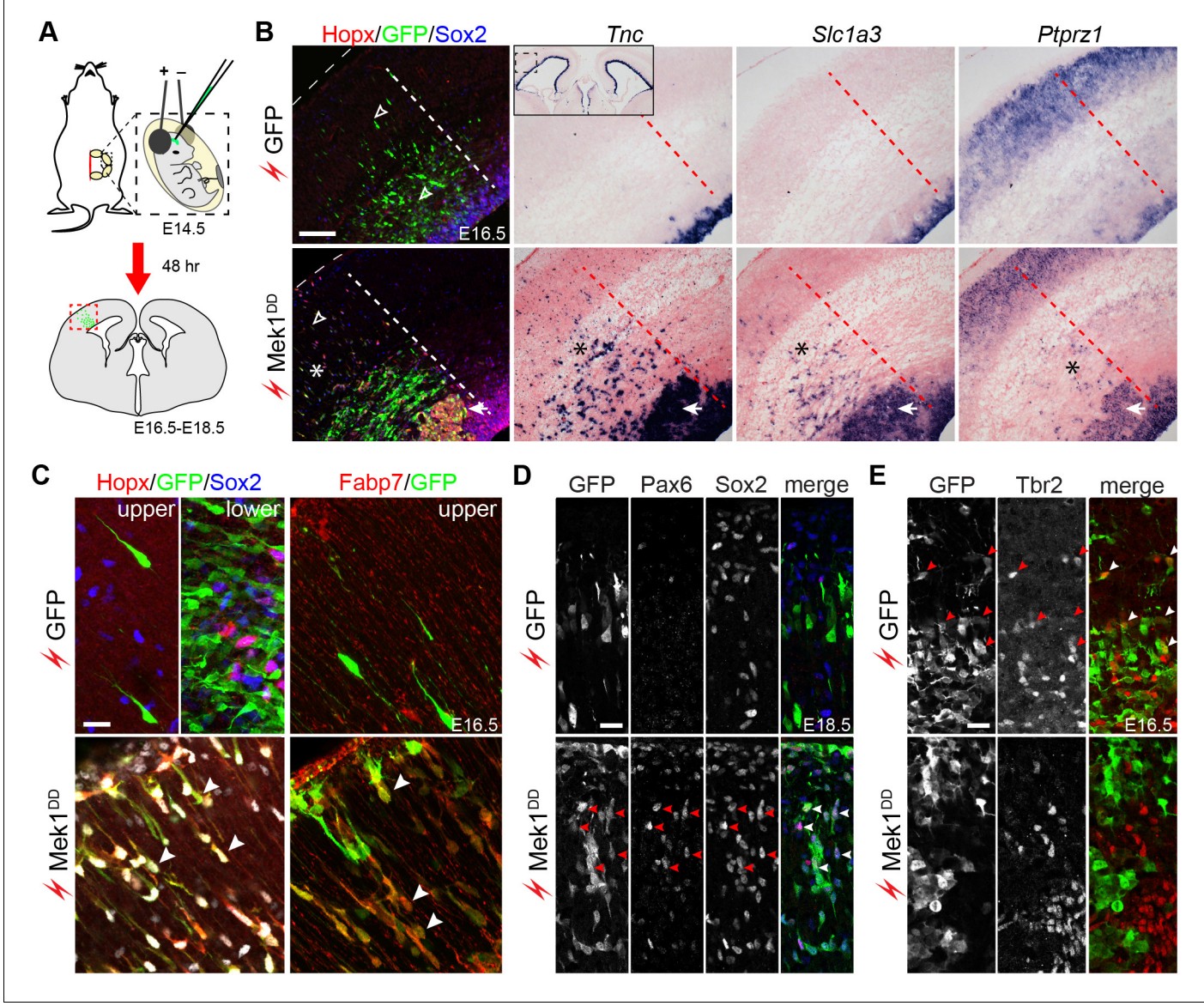

**Figure 5.** Hyperactivation of ERK signaling induces bRG in the mouse neocortex. (A) Schematic of the electroporation experiment. The box with the dotted lines indicates the approximate cortical area shown in the rest of the figure. (B–E) Immunofluorescence and in situ hybridization images of adjacent coronal sections of mouse cortices 48 hr (B, C, and E) and 96 hr (D) after the *in utero* electroporation of the indicated transgene. The thin dotted lines demarcate the pial surface; and the thick dotted lines separate the areas with and without electroporated cells; asterisks indicate the induction of bRG; the arrowheads in D indicate the heterotopia above the VZ; the inset in B is a low-magnification image showing *Tnc* expression in the cortical VZ on both the transfected and untransfected sides. A three-dimensional rendering of the induced bRG is shown in *Video 1*. Nuclei are stained with Hoechst 33342. Scale bars: 200 μm (B), 20 μm (C–E).

The following figure supplements are available for figure 5:

**Figure supplement 1.** *Mek1^DD*-induced bRG form heterotopia in the lower layer of the cortex.

**Figure supplement 2.** Electroporation of *EGFP* does not induce bRG.

occasionally caused the basal dispersion of aRG in the VZ (*Figure 5—figure supplement 2*), but only a small fraction of the dispersed GFP-expressing cells (3.3 ± 1.5%) expressed Hopx, whereas 31.8 ± 4.7% of the *Mek1^DD*-expressing cells (marked by GFP) simultaneously expressed Hopx and Sox2 at 48 hr after electroporation (*Figure 5B,C*, *Figure 5—figure supplement 1*, and *Video 1*).

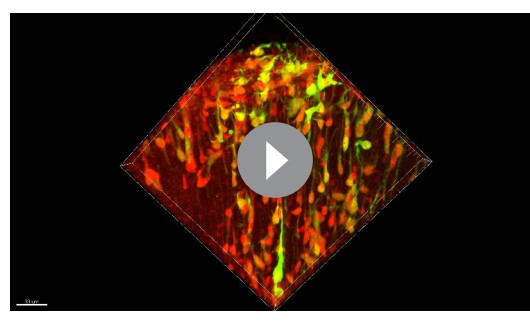

**Video 1.** Movie of three-dimension rendering of *Mek1^DD*-induced bRG in the mouse cortex. The red, green and blue channels represent antibody-staining for Hopx, GFP, and Sox2, respectively.

This showed that Mek1$^{DD}$ cell-autonomously induces bRG-like cells. Similar to the characteristic morphology of bRG in primate cortices (*Fietz et al., 2010*; *Hansen et al., 2010*; *Reillo et al., 2011*; *Kelava et al., 2012*; *Betizeau et al., 2013*; *Gertz et al., 2014*), the *Mek1^DD*-expressing cells that expressed bRG markers displayed long basal processes that reached the pial surface (78.2 ± 3.1%, n = 384) and occasionally also an apical process (21.8 ± 3.1%; *Figure 5C* and *Video 1*). In agreement with previous reports of an association between bRG and persistent *Pax6* expression (*Fietz et al., 2012*; *Wong et al., 2015*), 57.4 ± 3.8% of the *Mek1^DD*-expressing cells were positive for Pax6 (n = 1787 cells counted) in 48–96 hr after electroporation (*Figure 5D*). Notably, only 5.24 ± 1.75% *Mek1^DD*-expressing cells were positive for the IPC marker Eomes (n = 1,083; *Figure 5E*). Collectively, our data show that *Mek1^DD* expression induces bRG, but not IPC, in the mouse neocortex.

## Induced bRG undergo multiple rounds of self-renewing proliferation and give rise to upper-layer neurons

The proliferation and differentiation potential of the induced bRG in the mouse neocortex was studied using immunofluorescence. 48 hr after electroporation, most *Mek1^DD*-induced Hopx$^+$ cells were positive for Ki67 and a fraction of them was also positive for phospho-vimentin (pVim), a marker of M-phase cells (*Figure 6A*). This result demonstrated that most *Mek1^DD*-induced bRG were highly proliferative. To determine whether the induced bRG were capable to undergo multiple rounds of self-renewing divisions, a hallmark of basal progenitors in primate neocortex, embryos were electroporated at E14.5 and then treated with the thymidine analogues 5-bromodeoxyuridine (BrdU) and 5-ethynyldeoxyuridine (EdU) at E16.5 and E17.5, respectively (*Figure 6B*). In embryos electroporated with GFP, few GFP$^+$ cells were co-labeled with BrdU, EdU, or Ki67 at E18.5 (type 4 cells, *Figure 6B and C*). By contrast, most *Mek1^DD*-expressing cells were positive for at least two of the three labeling at E18.5 (types 1 and 2 cells, *Figure 6B and C*), consistent with their having undergone at least two rounds of divisions and/or continuing to divide 96 hr after electroporation. The latter conclusion was supported by the notably weakened GFP immunoreactivity in BrdU$^+$/EdU$^+$ cells (*Figure 6C*). Moreover, the clustering of *Mek1^DD*–expressing cells that were positive for Hopx, BrdU, and EdU between the SVZ and cortical plate (*Figure 6D*) suggested that Mek1$^{DD}$ induced bRG underwent clonal expansion and extensive self renewal.

To examine the mode of cell division of the induced bRG, we conducted clonal cell-pair assays (*Figure 6E*). This method has been extensively used to analyze symmetric versus asymmetric cell division in neural progenitor cells (*Shen et al., 2002*). We found that in transfected cells isolated from the cortex 48 hr after electroporation most *Mek1^DD*-expressing cells underwent symmetric division to generate two Fabp7$^+$ daughter cells (data not shown). At 72 hr after electroporation, all three possible cell division types were present among the GFP$^+$ *Mek1^DD*-expressing doublets: bRG-bRG (Fabp7-Fabp7), bRG-neuron (Fabp7-Tubb3), and neuron-neuron (Tubb3-Tubb3) (*Figure 6F and G*). Most of the *Mek1^DD*-expressing doublets underwent proliferative division (Fabp7-Fabp7) either from the upper (80.0%) or the lower (70.2%, including the SVZ and VZ) part of the cortex, in contrast to the predominantly neurogenic division (Tubb3-Tubb3, 64.5%) of the GFP-only-expressing doublets (*Figure 6G*). Collectively, our data show that expression of Mek1$^{DD}$ induces bRG-like cells that have extensive self-renewing and neurogenic potential.

To follow the fate of the induced bRG, markers of cortical neurons were examined in P3 cortices. In brains transfected with GFP alone, GFP$^+$ cells were arranged uniformly in the upper layer and expressed Satb2 (layer II-IV neurons), but not the deep-layer neuronal markers Tbr1 and Ctip2 (*Figure 7A* and data not shown). Many *Mek1^DD*-transfected cells with strong GFP also expressed Satb2 but they were arranged instead in clusters (*Figure 7A*), in agreement with the clonal

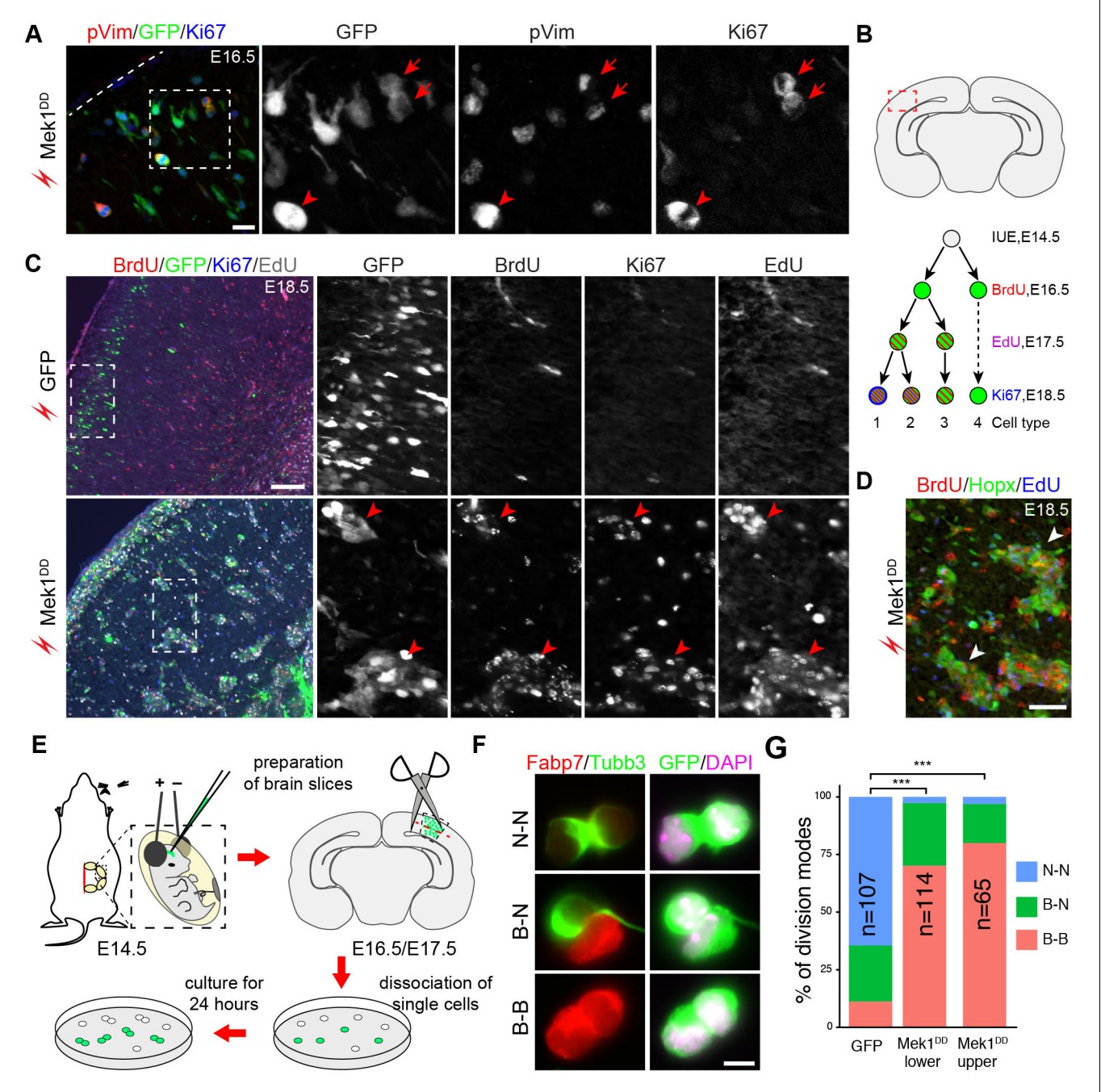

**Figure 6.** *Mek1DD*-induced bRG undergo multiple rounds of self-renewing divisions. (**A**) Immunofluorescence of phosphorylated vimentin (pVim), GFP, and Ki67 on coronal sections of E14.5 mouse cortex 48 hr after their electroporation. Dashed lines demarcate the pial surface; arrowheads indicate pVim-positive transfected cells (marked by GFP); arrows indicate a doublet of bRG at the end of mitosis. (**B**) Illustrations showing the boxed area that is corresponding to the images in this figure (upper) and the protocol used to study the proliferation of transfected cells (lower). BrdU/EdU/Ki67 triple labeling identifies four types of cells with distinct proliferation history. (**C and D**) Immunofluorescence on coronal sections of E18.5 mouse cortex electroporated *in utero* with *EGFP* or *Mek1DD* at E14.5. The boxed areas are enlarged and shown in individual channels. Arrowheads denote clusters of *Mek1DD*-transfected cells (**C**) and the induced Hopx-positive cells (**D**), which are mostly BrdU/EdU double (type 2) and BrdU/EdU/Ki67 triple (type 1) positive. (**E**) Illustration of the cell-pair assay procedure. The red dashed line indicates separation of the upper and lower parts of the cortex transfected with *Mek1DD*. (**F**) Immunocytochemistry of transfected cells (GFP+) and daughter-cell pairs. (**G**) Quantification of the percentage of transfected cells marked by GFP that underwent different modes of divisions. *P* values were calculated using a $\chi^2$ test. Scale bars: 20 µm (**A**), 200 µm (**C**), 50 µm (**D**), and 5 µm (**F**).

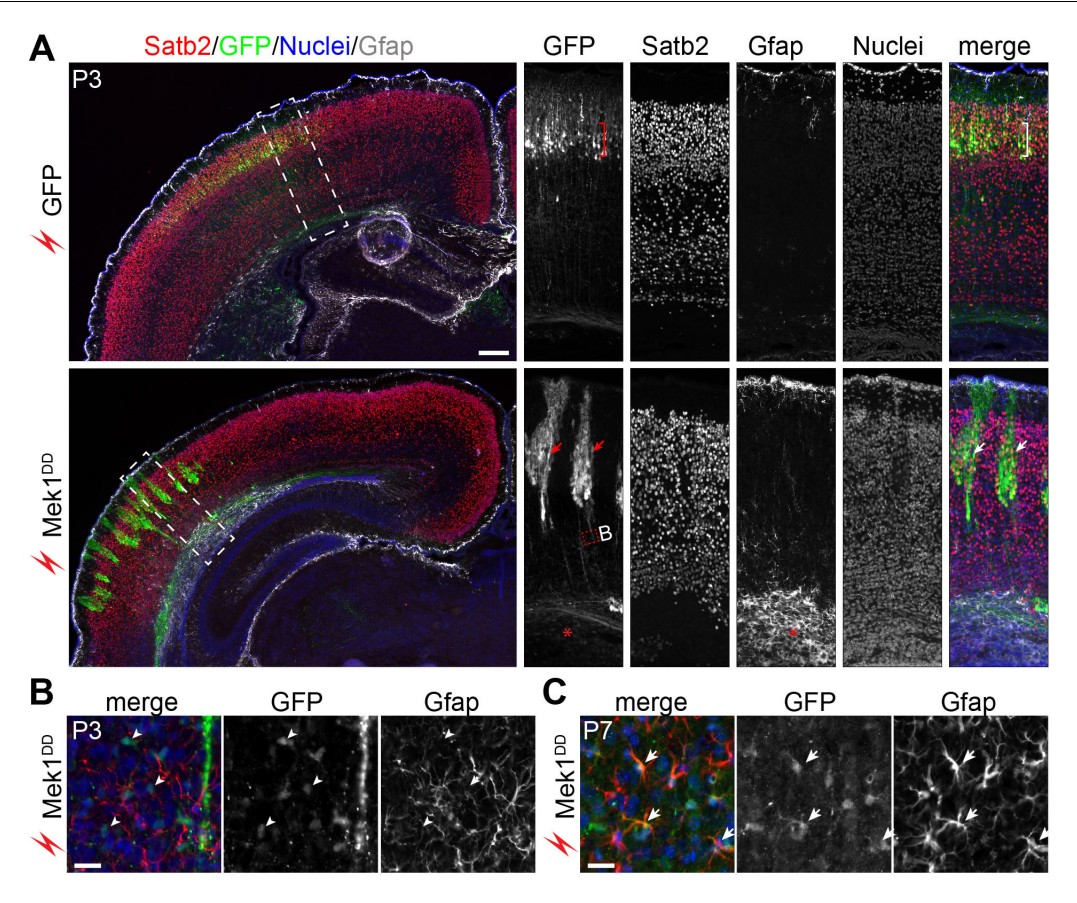

**Figure 7.** *Mek1$^{DD}$*-induced bRG form neurons and astrocytes. (**A**) Immunofluorescence of Satb2, GFP, and Gfap on coronal sections of mouse cortex 8 days after electroporation. The bracket indicates layers of Satb2-positive neurons derived from GFP-transfected cells; the arrows show clusters of Satb2-positive neurons formed by *Mek1$^{DD}$*-transfected cells; the asterisk denotes the accumulation of astrocytes; arrowheads indicate scattered *Mek1$^{DD}$*-transfected cells with weak GFP and negative for Gfap. (**B and C**) Immunofluorescence of Gfap and GFP on coronal sections of P3 (**B**) and P7 (**C**) brains electroporated at E14.5. Arrowheads and arrows point to absence and presence, respectively, of GFP in Gfap+ cells. Nuclei were stained with Hoechst 33342 and are shown in the blue channel.

expansion of the induced bRG observed at the earlier stages and the subsequent formation of Satb2$^+$ cells from the bRG. Although increased numbers of astrocytes marked by Gfap were detected in the area containing *Mek1$^{DD}$*-transfected cells compared to non-transfected area (*Figure 7A and B*), GFP-labeled *Mek1$^{DD}$*-tranfected cells in the P3 neocortex were mostly devoid of Gfap and the oligoglial markers Olig2 and Sox10 (*Figure 7B* and data not shown). At P7, however, some *Mek1$^{DD}$*-expressing cells with low-level GFP expression were positive for Gfap (*Figure 7C*), suggesting that the induced bRG are multipotent. Despite the ectopic bRG induction, *in utero* electroporation of *Mek1$^{DD}$* did not cause folding of the cortex (n = 10). Together, our data show that *Mek1$^{DD}$*-induced bRG have extensive proliferation capacity and give rise to both neurons and astrocytes. However, at least to the extent of increased number of bRG obtained with the electroporation folding of the mouse cortex is not induced by expansion of bRG in the absence of parallel expansion of IPCs.

## Induction of bRG and BG is similarly regulated by an FGF-ERK-ETV axis

We found that *ETV5* and, to a lesser degree, *Etv4* were increased in human cortical aRG compared to mouse aRG (*Figure 3C*). Etv4 and Etv5 are well-known effectors of the FGF-Ras-ERK signaling cascade (*Mao et al., 2009*; *Zhang et al., 2009*; *Hollenhorst et al., 2011*; *Li et al., 2012*; *Breunig et al., 2015*). Moreover, *ETV5* is specifically expressed in bRG among human cortical

progenitor cells (*Pollen et al., 2015*; *Thomsen et al., 2016*). Finally, the transcripts of *Etv4* and *Etv5* were missing in *Ptpn11*-cKO cerebella but were restored in *Ptpn11*-cKO;*Map2k1* cerebella at E14.5 (*Figure 8A*). These observations raised the question if an FGF-ERK-ETV cascade is involved in the genesis of bRG and BG.

To test this hypothesis, we first investigate the role of *Etv4* and *Etv5* in BG formation. Through tamoxifen-induced Cre-mediated recombination in $Gbx2^{creER/+}$;$R26^{Etv4DN/+}$ embryos at E9.5, we expressed a dominant negative Etv4 (Etv4$^{DN}$), a fusion between the Etv4 DNA-binding domain and the engrailed repressor domain (*Mao et al., 2009*), to block the redundant function of *Etv4* and *Etv5* in cerebellar aRG and their progeny. Fate-mapping study using a tdTomato red fluorescent protein (RFP) reporter $R26^{RFP}$ mouse line (*Madisen et al., 2010*), we found that the *Gbx2*-expressing cells at E9.5 gave rise to BG, Purkinje cells, GABAergic interneurons, and granule cell precursors (*Figure 8B* and *Figure 8—figure supplement 1*). Compared to cells that expressed RFP in $Gbx2^{creER/+}$;$R26^{RFP/+}$ embryos, significantly fewer Etv4$^{DN}$-expressing cells (marked by GFP) formed BG (*Figure 8B–D*). pERK immunoreactivity was detected in Etv4$^{DN}$-expressing cells in the cerebellar VZ at E13.5 (*Figure 8—figure supplement 2A*), indicating that ERK signaling acts upstream of Etv4 and Etv5. No difference in the expression of Olig2 (cerebellar GABAergic precursors) (*Seto et al., 2014*; *Ju et al.,*

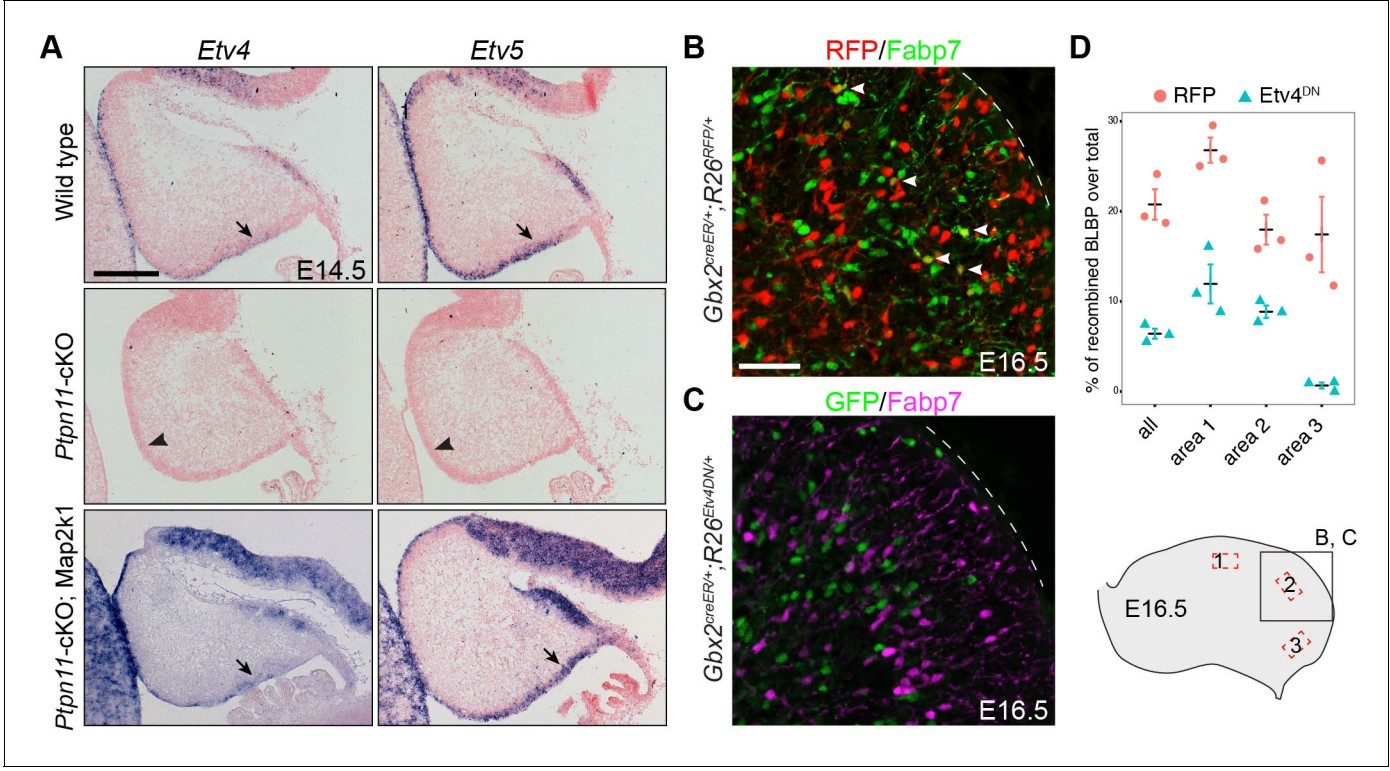

**Figure 8.** Etv4 and Etv5 are important for BG formation. (**A**) In situ hybridization for *Etv4* and *Etv5* on sagittal sections of E14.5 cerebella. The presence and absence of *Etv4* and *Etv5* transcripts are indicated by arrows and arrowheads, respectively. (**B and C**) Immunofluorescence on sagittal sections of E16.5 $Gbx2^{+/creER}$;$R26^{RFP/+}$ (**B**) and $Gbx2^{+/creER}$;$R26^{Etv4DN/+}$ cerebella (**C**) treated with tamoxifen at E9.5. (**D**) Quantification of *RFP* and *Etv4$^{DN}$*-expressing cells that display Fabp7 immunoreactivity in combined (all) or individual areas 1–3, as indicated in the illustration in the lower right corner. Unpaired Student's t-test, p=0.008268, $t_{(4)}$ = 8.059 (all); p=0.004507, $t_{(4)}$ = 5.760 (area 1); p=0.006968, $t_{(4)}$ = 5.103 (area 2); p=0.01634, $t_{(4)}$ = 3.985 (area 3).

The following figure supplements are available for figure 8:

**Figure supplement 1.** Contribution of *Gbx2*-expressing cells at E9.5 to different cerebellar cell types.

**Figure supplement 2.** Inactivation of Etv4 and Etv5 does not affect the activation of ERK, neurogenesis, or cell survival.

**Figure supplement 3.** Simultaneous deletion of *Fgfr1*, *Fgfr2*, and *Fgfr3* results in the similar phenotype as that found in *Ptpn11*-cKO mice.

2016a) was detected between *Etv4*$^{DN}$-positive and *Etv4*$^{DN}$-negative cells inside or near the cerebellar VZ in E13.5 *Gbx2*$^{creER/+}$;*R26*$^{Etv4DN/+}$ embryos (*Figure 8—figure supplement 2B*), suggesting that *Etv4*$^{DN}$ expression does not overtly alter neurogenesis from cerebellar aRG. Finally, we ruled out that Etv4$^{DN}$ caused cell death of developing BG by immunohistochemistry for activated caspase 3 (*Figure 8—figure supplement 2C*). Collectively, our data suggest that *Etv4* and *Etv5* are essential for BG formation.

Whether the forced expression of *Etv4* or *Etv5* restores BG in *Ptpn11*-cKO cerebella was investigated in a novel ex vivo electroporation procedure (*Figure 9A*). With this procedure, only cerebellar aRG, which line the ventricular zone surface and contact with DNA solution injected into the fourth ventricle, are transfected. In WT cerebellar slices after 24 or 48 hr of in vitro culture, aRG transfected with GFP formed BG, which were identified by markers (Fabp7 and Sox9) and the unipolar morphology of the cells, and non-BG cells (*Figure 9B*). In *Ptpn11*-cKO cerebellar slices, GFP-transfected cells were mostly restricted to the VZ, displaying long basal processes and immunoreactivity for Sox9 but not Fabp7 (*Figure 9B*). This result demonstrated that the inactivation of *Ptpn11* blocks the aRG-to-BG transition, as previously described (*Li et al., 2014*). Remarkably, electroporation of *Mek1*$^{DD}$ robustly rescued BG in *Ptpn11*-cKO cerebella (*Figure 9C*). Electroporation of *Etv4* or *Etv5* also

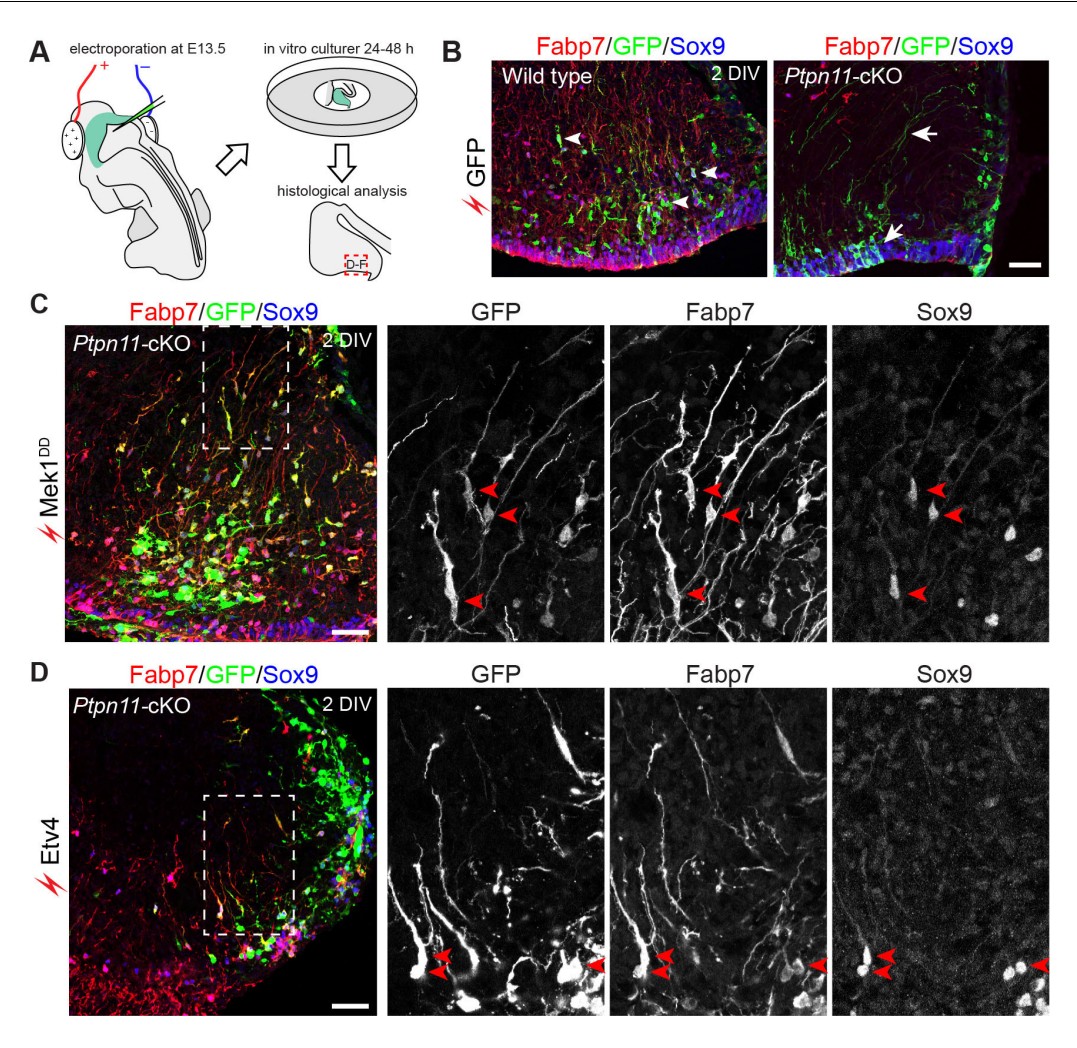

**Figure 9.** Activation of the ERK-ETV cascade in cerebellar aRG at E13.5 is critical to BG generation. (A) The procedure for ex vivo electroporation. (B–D) Immunofluorescence of Fabp7, GFP, and Sox9 on sections of cerebellar slices 48 hr (2 day in vitro, 2 DIV) after the electroporation of *EGFP* (B), *Mek1*$^{DD}$ (C), or *Etv4* (D). Arrows point to the rescued BG; boxed areas are enlarged and shown in individual channels. Scale bar: 50 μm.

rescued the formation of BG but to less extent compared to that of *Mek1*<sup>DD</sup> (*Figure 9D* and data not shown). The BG marker Fabp7 was selectively expressed in *Mek1*<sup>DD</sup>- or *Etv*-rescued BG in the cerebellar cortex, suggesting that Map2k1 and ETV act cell-autonomously to promote BG formation. Furthermore, inactivation of *Fgf9* or the three FGFR genes (*Fgfr1*, *2*, and *3*) depletes BG in the mouse cerebellum as found in *Ptpn11*-cKO mutants (*Lin et al., 2009*) (*Figure 8—figure supplement 3*), demonstrating the essential role of FGF signaling in BG formation. We did not attempt to rescue BG by FGF because Ptpn11 is essential to mediate FGF receptor (FGFR) signaling to ERK (*Hadari et al., 2001*). Collectively, these observations demonstrate that *Ptpn11* regulates the FGF-ERK-ETV cascade in the control of the aRG-to-BG transition.

Next, we examined whether the FGF-ERK-ETV cascade is important for bRG formation by *in utero* electroporation of constitutively active *FGFR1*, *FGFR1*<sup>K656E</sup> (*Olsen et al., 2006*), or *Etv4* in mouse cortex at E14.5. Similar to *Mek1*<sup>DD</sup>, *FGFR1*<sup>K656E</sup> and *Etv4* induced cells that expressed Hopx, Sox2, *Tnc*, *Slc1a3*, and *Ptprz1* 48 hr after electroporation (*Figure 10A*). Similar percentages of Hopx/Sox2 double labeling were found among *FGFR1*<sup>K656E</sup>-, *Mek1*<sup>DD</sup>-, and *Etv4*-expressing cells, and they were significantly higher than that in *GFP*- expressing cells (*Figure 10B*). These findings suggest that FGFR, Map2k1, and Etv4 act in the same pathway to induce bRG. Notable, *in utero* electroporation

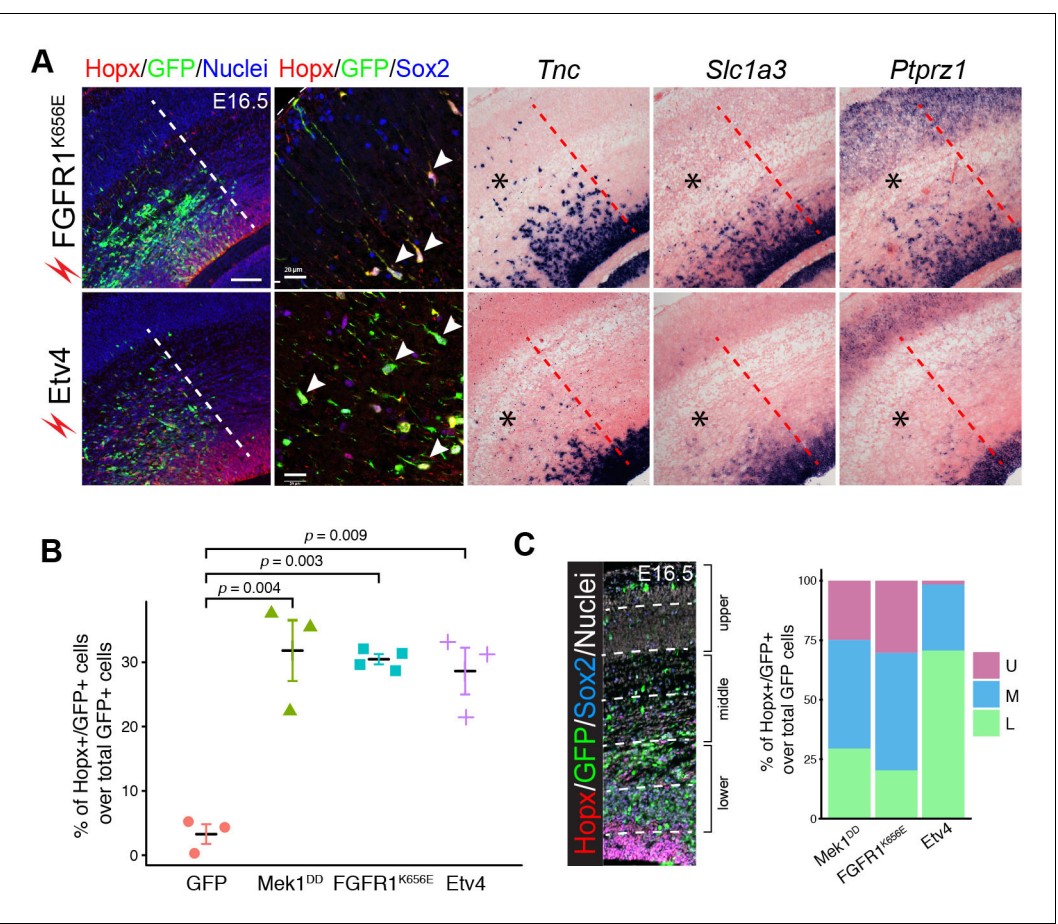

**Figure 10.** Expression of activated *FGFR* or *Etv4* induces bRG in the mouse neocortex. (**A**) Immunofluorescence and in situ hybridization images of adjacent coronal sections of mouse cortices 48 hr after the *in utero* electroporation of the indicated transgene. (**B**) Quantification of Hopx/GFP double-labeled cells relative to the total number of transfected cells. Each data point represents one embryo in which three or more adjacent sections were examined. Data are presented as the mean ± SEM. *P* values were calculated using a one-way ANOVA followed by a post-hoc Turkey-Kramer multiple comparison test, $F_{(9)}$ =21.76. (**C**) Distribution of Hopx-positive cells induced by *Mek1*<sup>DD</sup>, *FGFR1*<sup>K656E</sup>, and *Etv4* in the upper (**U**), middle (**M**), and lower (**L**) parts of the cortex. Scale bars: 200 μm.

of $Mek1^{DD}$, $FGFR1^{K656E}$, or $Etv4$ caused slightly different phenotypes, despite their similar function in inducing bRG-like cells. For example, the Hopx$^+$ cells induced by $Etv4$ occupied a more apical position than those induced by $FGFR1^{K656E}$ and $Mek1^{DD}$ (**Figure 10C**). Furthermore, $Mek1^{DD}$, but not $FGFR1^{K656E}$ and $Etv4$, induced heterotopia near the VZ, and these heterotopia were composed of Sabt2+ cortical neurons with few Gfap+ astrocytes after birth (**Figure 5B** and *Figure 5—figure supplement 1*). Taken together, our results show that activation of the FGF-ERK-ETV axis is involved in inducing the transition of aRG to BG in the cerebellum and to bRG in the neocortex.

## Discussion

In the present study, we investigated the similarities between bRG and BG in their genome-wide transcriptional profiles and developmental programs. Using differential gene expression and coexpression studies based on RNA-seq, we establish the molecular features of mouse nascent BG, and show the remarkably similarity between human bRG and mouse BG in their molecular characteristics. Through transcriptome analysis and immunohistochemistry, we demonstrate that heightened ERK signaling is associated with the timely formation of endogenous bRG in the mouse cortex, while stronger ERK activity in cortical aRG is linked to the abundance of bRG in the human compared to mouse cortex. We show that the activation of the FGF-ERK-ETV axis that is important for BG formation induces multipotent bRG, which display canonical human bRG gene markers and extensive proliferative capacity, in the mouse neocortex. Our results have revealed surprising parallels in the transition of aRG to bRG in the neocortex, and aRG to BG in the cerebellum.

By taking advantage of the specific loss of BG in $Ptpn11$-cKO and the rescue of BG in $Ptpn11$-cKO;$Map2k1$ cerebella, we used RNA-seq to examine the transcriptome of the cerebella with and without BG at the stages before and after BG induction. Using conventional differential expression analysis and WGCNA, we identified putative markers for nascent BG. The validation of these markers was mostly carried out by inspection of expression data of Allen Mouse Brain. Because of the inherent difficulty to assign cell-type specific expression based on in situ hybridization, the gene expression analysis was performed by an examiner who was blinded to the gene symbol and the origin of the gene list. We showed that candidate gene lists that were produced by the two statistical methods were both significantly enriched in BG in P56 mouse cerebellum compared to randomly generated gene lists. One potential caveat is that BG gene expression may be dynamic during development. However, interrogation of BG-specific expression in the Allen Mouse Developmental Brain and GenePaint has revealed that many BG markers, such as $Fabp7$, $Hopx$, $Etv5$, $Ptprz1$, $Slc1a3$ and $Tnc$ are expressed in both nascent and mature BG (*Figure 1—figure supplement 1A*). Another caveat is that our strategy would likely identify downstream targets of the Ptpn11-ERK pathway, which may also be involved in development of non-BG cells in the cerebellum. Single-cell RNA-seq analysis will help clarify the definite markers for nascent BG.

$Ptpn11$ and other components of the FGF-ERK pathway play a multifaceted roles in the progression of neural progenitors, including the timely transition from neuroepithelial cells to aRG (*Sahara and O'Leary, 2009*), from aRG to IPC (*Kang et al., 2009*), from proliferative to neurogenic self-renewing aRG (*Dee et al., 2016*), and from neurogenic to gliogenic aRG (*Gauthier et al., 2007*; *Ke et al., 2007*). After their broad expression in the cerebellar primordium at E10.5, the transcripts of $Etv4$ and $Etv5$ disappear, except for the region near the mid-hindbrain junction, by E12.5 (*Li et al., 2014*), but re-appear in cerebellar aRG at E14.5 (**Figure 8A**), following the robust expression of pERK in these cells at E13.5 (**Figure 4D**). While $Ptpn11$ deletion does not change the transcription of $Etv4$ and $Etv5$ at E10.5 (*Li et al., 2014*), neither gene is expressed in the cerebellar anlage of $Ptpn11$-cKO embryos after E12.5 (**Figure 8A**). This suggests that activation of ERK-ETV in cerebellar aRG at E13.5 but not before is essential for BG induction. In the current study, we have extended our previous findings by showing that the expression of $Mek1^{DD}$, $Etv4$, or $Etv5$ cell-autonomously rescues BG in $Ptpn11$-cKO cerebella at E13.5. Coincidentally, strong ERK signaling is present in the VZ starting at E13.5 to at least E16.5, matching closely with the onset and duration of BG generation from the VZ. These results demonstrate that heightened ERK signaling activity in cerebellar aRG controls the timely aRG-to-BG transition. In parallel with the BG induction, intensified pERK in cortical aRG correlates with the formation of endogenous bRG in the mouse cortex at E17.5 (**Figure 4E–I**), whereas the hyperactivation of the ERK signaling induces ectopic bRG at E14.5 (**Figures 5** and **10**). Therefore, the intensification of ERK signaling controls the timely transition of aRG

to BG in the cerebellum, and to bRG in the neocortex. How ERK signaling controls the formation of BG and bRG is currently unknown. Previous studies have shown that increasing the horizontal division, in which the cleavage furrow is parallel to the ventricular surface, contributes to the generation of bRG in both the human and mouse cortex (*LaMonica et al., 2013*; *Martínez-Martínez et al., 2016*). As it has been shown that the ERK pathway determines mitotic spindle orientation epithelial cells (*Tang et al., 2011*), it is possible that ERK signaling controls the generation of BG and bRG by regulating the spindle orientation. On the other hand, the transformation of aRG into bRG resembles the epithelial-to-mesenchymal transition (*Itoh et al., 2013*; *Pollen et al., 2015*), a developmental process that is regulated by the FGF-ERK pathway (*Lamouille et al., 2014*). Future studies will be needed to determine whether FGF-ERK signaling promotes the transition from aRG to BG or bRG similarly to the regulation of the epithelial-to-mesenchymal transition, and/or by regulating spindle orientation.

A previous study showed that *in utero* electroporation of *Mek1^{DD}* or *Etv5* in E15.5 mouse neocortex enhances the generation of proliferative Fabp7$^+$ cells, but no other bRG markers were examined (*Li et al., 2012*). The authors concluded that these Fabp7$^+$ cells are astrocyte precursors because 20–25% of the transfectants become astrocytes at P22 and P60 (*Li et al., 2012*). Although we detected supernumerary Gfap+ cells in cortical region with *Mek1^{DD}* transfection, many of the Gfap+ astrocytes were not colocalized with EGFP, which marked *Mek1^{DD}* transfection, at P3 (*Figure 7B,C*, and *Figure 5—figure supplement 1*). Because EGFP expression was rapidly diminished during the successive division of the induced bRG, we were uncertain whether the transfection of *Mek1^{DD}* cell autonomously or non-autonomously induced astrocytes. In the current study, we showed that forced expression of *Mek1^{DD}* induced neurogenic progenitor with remarkably high proliferation and self-renewing potential (*Figure 6*). Furthermore, these progenitors express many canonical human bRG markers (*Pollen et al., 2015*; *Thomsen et al., 2016*) (*Figures 5* and *10*). Importantly, Pax6, which is not linked to astroglial lineage, was maintained in *Mek1^{DD}*-induced bRG at least to E18.5 (*Figure 5D*). In basal progenitors (including IPCs and bRG), persistent Pax6 expression is a hallmark of the primate neocortex (*Mo and Zecevic, 2008*; *Fietz et al., 2010*; *Hansen et al., 2010*; *Reillo et al., 2011*; *Betizeau et al., 2013*). Indeed, sustained Pax6 expression is sufficient to expand IPC and bRG-like cells in the mouse neocortex (*Wong et al., 2015*). Importantly, we show that the expression of *Mek1^{DD}* primarily induced bRG, whereas IPC marker Eomes was mostly absent from *Mek1^{DD}*-expressing cells (*Figure 5*). Although the expansion of bRG-like cells and IPC through various genetic manipulations has previously been reported, to the best of our knowledge, this is the first report of the specific induction of primate-like bRG in the mouse neocortex. Together, our data strongly suggest that activation of the FGF-ERK-ETV cascade is involved in the induction of bRG in the mouse neocortex.

In any given species, the relative abundance of bRG can be attributed, at least partially, to the bRG-forming competence of aRG and/or to the self-expansion potential of bRG. We demonstrated that FGF-ERK-ETV activity is lower in the mouse than in the human neocortical aRG (*Figures 3* and *4*), and that activation of this axis is sufficient to induce bRG (*Figures 5–7* and *10*). These observations indicate that a higher level of FGF-ERK-ETV activity in aRG contributes to the greater abundance of bRG in humans than in mice. Comparative epigenetic profiling of human, rhesus macaque, and mouse corticogenesis has shown that *cis*-regulatory elements with activity in humans are enriched in the modules of coexpressed genes belonging to the FGF and TGF$\beta$ pathways (*Reilly et al., 2015*). In agreement with these findings, our coexpression network analysis revealed significant divergence in FGF-ERK and BMP signaling pathways in human and mouse corticogenesis (*Figure 3F*). Therefore, gene regulation changes that are specific to the human lineage modify corticogenesis in humans, in part by enhancing FGF-ERR signaling activity in aRG leading to expansion of bRG. Gain-of-function mutations of *FGFR2* and *FGFR3* cause Apert syndrome and thanatophoric dysplasia, which are characterized by unique and complex malformation of the cortex, including megalencephaly and polymicrogyria (*Hevner, 2005*). Moreover, mutations of the genes of the ERK signaling cascade have been implicated in neuro-cardio-facial-cutaneous (NCFC) syndromes that include an abnormal size and gyrification of the neocortex and cerebellum (*Samuels et al., 2009*). Our new findings imply that abnormal bRG development may contribute to the pathologies of the human neocortex that are caused by aberrant FGF-ERK function.

Despite the similarity in their generation, BG do not produce neurons during normal development (*Parmigiani et al., 2015*), unlike bRG in the human cortex. Although the *Mek1^{DD}*-induced bRG

possess extensive proliferative self-renewing potential, no dramatic increase in cortical neurons was observed in neocortex transfected with $Mek1^{DD}$. Furthermore, both the endogenous and Mek1$^{DD}$-induced bRG are distributed in both the SVZ and the cortical plate in the mouse cortex rather than being restricted to the outer SVZ as described in the primate cortex. Therefore, additional intrinsic and/or extrinsic factors may be required for the neurogenic potential and localization of human bRG. Comparative studies in human, ferret, and mouse have shown that the neocortical basal progenitors of humans exhibit a greater neuronal lineage commitment and degree of differentiation, partially through the activity of proneural genes (*Johnson et al., 2009*). In vitro and in vivo studies have shown that the forced expression of proneural genes, such as *Ascl1*, *Neurog2*, and *NeuroD1*, can reprogram glial cells to neocortical neurons (*Guo et al., 2014*; *Masserdotti et al., 2015*; *Zhang et al., 2015*). Further studies are required to determine whether the expression of proneural genes enhances the neurogenic potential of BG in the cerebellum or $Mek1^{DD}$-induced bRG in the mouse neocortex.

The expression of $Mek1^{DD}$ specifically expands bRG but fails to induce folding of the mouse neocortex. These findings are in agreement with the notion that an abundance of bRG is necessary, but insufficient, for gyrencephaly (*Hevner and Haydar, 2012*; *Kelava et al., 2012*). Rather, increased neuronal output through the expansion of other basal progenitors together with bRG may be necessary for folding of the neocortex. A notable parallel may be cerebellar foliation, in which the important role played by the proliferation of granule cell precursors has been well documented. For example, functional alterations of Shh, which promotes granule cell precursor proliferation in the EGL (*Dahmane and Ruiz i Altaba, 1999*; *Wallace, 1999*; *Wechsler-Reya and Scott, 1999*), have been linked to the extent of cerebellar foliation (*Corrales et al., 2004*, *2006*). Interestingly, hedgehog signaling also promotes the development of BG (*Dahmane and Ruiz i Altaba, 1999*; *Fleming et al., 2013*) and bRG (*Wang et al., 2016*). These observations suggest that the hedgehog and FGF-ERK signaling pathways act in concert to promote neuronal output via IPC or granule cell precursors as well as to expand bRG and BG populations in the evolution of a convoluted neocortical and cerebellar cortex, respectively.

## Materials and methods

### Mouse and tissue preparation

Husbandry of mice was carried out according to guidelines approved by the University of Connecticut. Light/dark cycle in the vivarium was 12 hr light on and 12 hr light off. All mouse strains were maintained on CD-1 outbred genetic background. The day of vaginal plug detection was considered embryonic day (E) 0.5. For tamoxifen administration, 4–6 milligrams of tamoxifen (Sigma, St. Louis, MI) in corn oil were administered to pregnant females through oral gavage as described (*Li and Joyner, 2001*). Generation and characterization of the $En1^{cre}$ ($En1^{tm2(cre)Wrst}$/J; #007916) (*Li et al., 2002*), $Gbx2^{creER}$ ($Gbx2^{tm1.1(cre/ERT2)Jyhl}$/J; #022135) (*Chen et al., 2009*), $Ptpn11^{floxed}$ (*Yang et al., 2013*), $R26^{Etv4DN}$ (*Mao et al., 2009*), and $R26^{Mek1DD}$ (Gt(ROSA)$^{26Sortm8(Map2k1*,EGFP)Rsky}$/J; #012352) (*Srinivasan et al., 2009*) alleles have been previously reported. The $R26^{Etv4DN}$ allele contained the $Etv4^{DN}$-ires-YFP bicistronic sequence downstream of a *loxP*-flanked *Neo-STOP* cassette. Therefore, $Etv4^{DN}$-expressing cells and their progeny were permanently marked by YFP (recognized by anti-GFP antibodies) after tamoxifen-induced creER activation.

Embryonic mouse brains were dissected in cold phosphate buffered saline and fixed in 4% paraformaldehyde for 40 min to overnight. Brains were cryoprotected, frozen in Tissue-Plus (Thermo-Fisher Scientific, Carlsbad, CA), and sectioned in a cryostat (Leica, Germany, CM3050S).

### Histochemistry, immunofluorescence, and in situ hybridization

Standard protocols were used for immunofluorescence and in situ hybridization as described (*Chen et al., 2009*). Detailed protocols are available on the Li Laboratory website (http://lilab.uchc.edu/protocols/index.html). As the antibodies for Hopx and pERK were both raised in rabbits, we used a two-step technique as described previously (*Shindler and Roth, 1996*). First we used the sensitive Tyramide Signal Amplification Kits (ThermoFisher Scientific) to detect a highly diluted anti-pERK antibody. Subsequently, conventional immunostaining was performed to detect Hopx. We confirmed that the conventional method could not detect the bound anti-pERK antibodies in the

earlier steps. Primary and secondary antibodies used in the study are listed in the *Supplementary file 5A*.

To generate riboprobes for RNA in situ hybridization, PCR primers were designed using Primer 3 (*Untergasser et al., 2012*) to amplify a 500–700 bp region of the open reading frame for a given gene; a T7 promoter sequence appended to the reverse primer enabled direct generation of anti-sense riboprobes from T7-mediated transcription of PCR products. Primer sequences are listed in *Supplementary file 5B* and *Supplementary file 5C*. To generate cDNA, total RNA was extracted from E13.5 mouse brain tissues using Trizol extraction kit (Invitrogen, Carlsbad, CA) and reverse transcribed with Superscript III First Strand Synthesis System using random hexamers (Invitrogen).

## RNA sequencing and data analysis

The cerebellar anlage was microdissected from E12.5 and E13.5 brains. For E14.5 cerebellar anlage, microdissection was performed on brain slices (300 μm thickness) that were prepared with a vibratome (Leica, VT1000S). Total RNA was isolated with TRIzol (Invitrogen) or Maxwell 16 LEV RNA FFPE Kit for automated RNA isolation (Promega, Madison, WI). Approximately 500 ng of total RNA, with a RNA integrity number of at least 7.5 (mostly above 9.0), was used for library preparation with Illumina TruSeq RNA Sample Prep Kit v2 (for E12.5 and E14.5 samples) or TrueSeq Stranded mRNA LT (E13.5). For the E12.5 and E14.5 samples, eight libraries were sequenced in two lanes with HiSeq2000 (Illumina, San Diego, CA) using 50-base single end sequencing chemistry. For the E13.5 samples, 21 of them were pooled and run on NextSeq500 (Illumina) high output flow cell using 75-cycle single end sequencing chemistry. RNA-seq raw data have been deposited in the Gene Expression Omnibus under accession codes GSE87104.

For expression quantification, RNA-seq data were mapped to reference genome sequences of mouse (GRCm38) and human (GRCh38) with STAR version 2.42a (*Dobin et al., 2013*). Gene annotation files from GENCODE (*Harrow et al., 2006*) were used for mouse (vM5) and human (V22). Resulted BAM files were used to generate gene counts with *featureCount* (*Liao et al., 2014*) using the uniq-counting mode.

Differential expression analysis was performed by *DESeq2* (*Love et al., 2014*). Batch effects from library preparation and sequencing were removed using the *ruv* R package (*Risso et al., 2014*). Pathway and functional analysis of gene lists were performed using the Go-Elite (*Zambon et al., 2012*).

## Comparison of gene expression in neocortical aRG of human and mouse

To examine gene expression in aRG in human and mouse neocortex, FASTQ files were obtained using the *fastq-dump* program of the SRA toolkit from the Gene Expression Omnibus, under the accession numbers GSE30765, GSE38805, GSE65000, and GSE66217. These datasets contained 13 human and 21 mouse RNA-seq samples of the VZ or aRG. Sequencing reads mapping, expression quantification and differentiation analysis were performed as described above. Orthologous genes were downloaded from Ensembl release 85. Human-mouse orthologs were defined as single-copy genes conserved in human and mouse. Because of different efficiency of mRNA enrichment among the different studies, rRNA and mitochondria genes were excluded resulting in a total 16,036 features (Ensembl ID). To assess the overall similarity between samples, we first transformed the count data into gene expression matrix using the *varianceStabilizingTransformationby* function of *DESeq2* (*Love et al., 2014*), and used the R function *dist* to determine the Euclidean distance between samples (*Figure 3—figure supplement 1A*) and the *plotPCA* function of *DESeq2* to show the sample relationship (*Figure 3—figure supplement 1B*).

To perform GSEA (*Subramanian et al., 2005*), expression matrix was generated by variance stabilizing transformation of the RNA-seq count data using the *DESeq2* package (*Love et al., 2014*). A comprehensive and monthly-updated gene-set containing all mouse pathways was downloaded from Bader lab (http://download.baderlab.org/EM_Genesets/current_release/) (October_01_2016 version). We added the BG-specific genes and ERK responding genes to the gene lists. GSEA was done using gene shuffling for P value estimated with 1000 permutations. Results of GSEA are listed in in *Supplementary file 4A*.

## Microarray analysis

Microarray data of time course study of ERK activation (*Hamilton and Brickman, 2014*) were retrieved from the Gene Expression Omnibus using *GEOquery* (*Davis and Meltzer, 2007*) under the accession number GSE59755. Technical sources of variation were removed with *ComBat* function of the *sva* package (*Leek and Storey, 2007*). Differentially expressed genes with two fold changes at different time points relating to time 0 was identified with the *limma* package (*Ritchie et al., 2015*). Results of differential gene expression analysis are shown in *Supplementary file 3*.

## Validation of gene expression in cerebellar BG

Expression data from E13.5 to P56 were automatically batch downloaded from the Image Download Service of Allen Brain Institute. BG-specific expression was manually confirmed using the following criteria. As BG are interlocked with Purkinje cells in the Purkinje cell layer (PCL), genes without any detectable signals in the PCL were scored as 0. For those with signals in PCL, genes that are specific to Purkinje cells, which were identified by their bigger, round and distinct soma, were scored as 1, those with indistinguishable between BG and Purkinje cells as 2, those that are specific to BG, which were identified by their smaller and irregular soma, as well as their radial projections around the Purkinje cells into the molecular cell layer, as 3 (*Figure 1—figure supplement 2*). Scoring was performed by an examiner blinded to gene symbols. Fisher's exact test was performed to determine if BG-specific genes were significantly enriched in the bRG- and BG-specific gene lists compared with the randomly selected gene of comparable numbers.

## Weighted gene Co-expression network analysis

Gene expression matrix was generated by variance stabilizing transformation of the RNA-seq count data using the *DESeq2* package (*Love et al., 2014*). Expression data were normalized for batch effect (*Leek, 2014*) and outlier removal (Z.K less than –2) using *SampleNetwork* R function (*Oldham et al., 2008*). From these processed expression data, we followed the protocols of WGCNA (*Zhang and Horvath, 2005*) to create a gene co-expression network. Modules were defined as branches of a hierarchical cluster tree using the top-down dynamic tree cut method (*Langfelder et al., 2008*). For each module, the expression patterns were summarized by the module eigengene (ME), defined as the singular vector of the standardized expression patterns. Pairs of modules with high module eigengene correlations ($R > 0.85$) were merged. The module membership ($k_{ME}$) for each gene with respect to each module was then defined as the Pearson correlation between the expression level of the gene and the module eigengene (*Oldham et al., 2008*).

To study the preservation of human and mouse cortical coexpression networks, we combined available RNA-seq data from all cortical cell types (*Ayoub et al., 2011*; *Fietz et al., 2012*; *Florio et al., 2015*; *Johnson et al., 2015*). After careful filtering and preprocessing of the data to remove batch effects and outliers (*Oldham et al., 2012*), 37 human (WG13-18) and 40 mouse (E14.5) samples were included. Between the human and mouse datasets, both gene expression and connectivity were highly preserved ($R = 0.78$, $p<1 \times 10^{-200}$ for expression; $R = 0.33$, $p<1 \times 10^{-200}$ for connectivity), indicating that the cross-species datasets are well matched. We used the *modulePreservation* function in R (*Langfelder et al., 2011*) to study the preservation of 11 signaling pathways (KEGG and Reactome), pan-RG signature genes (*Lui et al., 2014*), early-response genes induced by ERK, and signature genes for aRG, bRG, and IPC. The complete results are shown in *Supplementary file 4B*.

## In utero and ex vivo electroporation, and culture of brain slices

*In utero* electroporation was performed as described previously (*Shimogori and Ogawa, 2008*). Briefly, 1–2 µl of plasmid DNA (1.5 µg/µl) were injected into the lateral ventricles of E14.5 brains and electroporated using four pulses at 40 V for 50 ms at 100 ms intervals through the uterine wall using a BTX ElectroSquarePorator (BTX, Holliston, MA, ECM 830).

Ex vivo electroporation was performed in E13.5 embryos. Briefly, embryos were placed in ice cold Krebs buffer containing 126 mM NaCl, 2.5 mM KCl, 1.2 mM $NaH_2PO_4$, 1.2 mM $MgCl_2$, 2.5 mM $CaCl_2$, 11 mM glucose, and 25 mM $NaHCO_4$. *Ptpn11*-cKO embryos were identified based on truncation of the tectum (*Li et al., 2014*), and DNA solution was injected into the VI ventricle. To increase the efficiency of the experiment, both sides of the cerebellar anlagen were electroporated by

rotating the orientation of the electrode using five pulses at 60 V for 50 ms at 100 ms intervals. After electroporation, brains were dissected and embedded in 4% low-melting agarose (Seakem, VWR International, Radnor, PA). Sagittal brain slices in 300 µm thickness were prepared on a vibratome (Leica, VT1000S), and kept in cold Krebs buffer on ice, and the sections were transferred to serum medium (Invitrogen, MEM with glutamine, 10% fetal calf serum, 0.5% glucose and penicillin/strepto-mycin antibiotics). After 15 min, the sections were transferred to polycarbonate culture membranes (Whatman™ 13mm Nuclepore™, Fisher Scientific) in Falcon organ tissue culture dishes containing 1 ml of Neurobasal/B-27 medium (Neurobasal with 1x glutamine, 1% B-27, 0.5% glucose and penicil-lin/streptomycin antibiotics). They were subsequently incubated at 5% $CO_2$ and 37°C for 48 hr. After incubation, slices were fixed in 4% paraformaldehyde/phosphate buffered saline, washed in phos-phate buffered saline, embedded in Tissue-Plus (ThermoFisher Scientific), and sectioned in a Cryo-stat (Leica, CM3050S). The sections were subjected to standard in situ hybridization and immunofluorescence procedures.

### Expression construct

The full length cDNA for *Mek1$^{DD}$*, *Etv4*, *Etv5*, and *FGFR1K$^{656E}$* was cloned into the *pMES* expression vector (*Xiong et al., 2009*), placing upstream of an internal ribosomal entry site (*ires*) and the cDNA encoding enhance green fluorescent protein (*EGFP*). The expression cassette is under the control of cytomegalovirus early enhancer/chicken $\beta$-actin (CAG) promoters.

### Pair-cell assays

Pair-cell analysis was performed as described (*Shen et al., 2002*). Following electroporation at E14.5 as described above, electroporated brains at E16.5 and E17.5 were sectioned using a vibratome (Leica, VT1000S). Cortical tissues that were enriched for transferred cells (marked by GFP) were dis-sected with tungsten needles from cortical sections. For the cortex transfected with *Mek1$^{DD}$*, the upper and lower half of the cortex were further separated. To dissociate cells, dissected tissues were incubated in a protease solution containing 10 unit/ml papain (Fluka, Japan), 1000 units/ml DNAse I (Roche, Switzerland) and 5 mM L-cysteine in DMEM (Invitrogen), and triturated using a fire-polished Pasteur pipette to create a single-cell suspension. Cells were resuspended in culture medium con-taining DMEM, glutamine, penicillin/streptomycin, sodium pyruvate (Invitrogen), 1 mM N-acetyl-L-cysteine (Sigma), B27, N2 and 10 ng/ml bFGF2 (Invitrogen) and plated onto coverslips coated with poly-L-lysine (Sigma) at clonal density. The cultures were maintained in a humidified incubator at 37°C with constant 5% $CO_2$ supply. In 24 or 48 hr later, the cultures were fixed and immunostained for GFP, Fabp7, and Tubb3 and counterstained with the DNA dye Hoechst 33342 solution (Invitrogen).

### Quantifications and statistical analysis

To determine the co-localization of markers, we divided the dorsolateral cortex into six bins paralle-ling to the ventricular surface with the sixth bin immediate above the ventricular zone. Images of each bin were automatically produced by ImageJ software, and the images were randomized and coded. The number of Sox2$^+$ nuclei in each bin was counted with ImageJ. The numbers of GFP$^+$ and GFP$^+$/Hopx$^+$ cells in each bin were counted manually with the examiner blinded to relevant varia-bles, such as DNA constructs and bin numbers.

Data processing, statistical analysis and plotting were performed in R version 3.2.5. An unpaired two-tailed t-test with Welch's correction or Student's t-test was used for analysis of experiments involved two groups. One-way ANOVA followed by Turkey-Kramer multiple comparison test was used for analysis of experiments involving more than two groups with one comparison. Bartlett's test was performed to verify equal variance assumption before ANOVA and Student's t-test. Repro-ducible results were obtained from three or more samples, and quantitative data are expressed as means ± standard error of the mean (SEM).

## Acknowledgements

We thank Drs. Nada Zecevic and John Wizeman for critical reading and comments of the manuscript. We also thank Dr. Justin Cotney for his advices on RNA-seq and coexpression network analyses. We are grateful to Dr. Nada Zecevic for providing the human fetal brain tissues, Dr. Xin Zhang for

providing full-length cDNAs of mouse *Etv4* and *Etv5*, Dr. Aimin Liu for providing the human *FGFR1 (K656E)* cDNA, and Dr. Eric Turner for providing *pMES* expressing vector; Dr. Andrew McMahon for the *R26^{Etv4DN}* mice; Dr. Rashmi Bansal for the *R26^{Mek1DD}* mice. The monoclonal anti-NEFM antibody (2H3) was developed by TM Jessell and J Dodd, and obtained through the Developmental Studies Hybridoma Bank under the auspices of the NICHD and maintained by The University of Iowa (Iowa City, IA).

## Additional information

### Funding

| Funder | Grant reference number | Author |
| --- | --- | --- |
| National Institutes of Health | R01MH094914 | James YH Li |

The funders had no role in study design, data collection and interpretation, or the decision to submit the work for publication.

### Author contributions

XH, QG, Data curation, Formal analysis; AWL, Data curation, Formal analysis, Writing—review and editing; JYHL, Conceptualization, Supervision, Funding acquisition, Writing—original draft, Writing—review and editing

### Author ORCIDs

James YH Li, http://orcid.org/0000-0002-9231-2698

### Ethics

Animal experimentation: This study was performed in strict accordance with the recommendations in the Guide for the Care and Use of Laboratory Animals of the National Institutes of Health. All of the animals were handled according to approved institutional animal care and use committee (IACUC) protocols (101159-0918) of the University of Connecticut Health. All surgery was performed under sodium pentobarbital anesthesia, and every effort was made to minimize suffering.

## Additional files

### Supplementary files

• Supplementary file 1. Marker genes for human neocortical aRG and bRG. (A) Summary of consensus markers (first spreadsheet) based on two single-cell RNA-seq datasets (*Pollen et al., 2015*; *Thomsen et al., 2016*). (B) Table listing putative BG-specific expression of the orthologs of human bRG markers (second spreadsheet). Column A and B: human and mouse gene symbol, respectively; column C: bRG markers identified by SINCERA and/or Pollen et al; column D: scores of BG specific expression; column E and F: BG-specific genes identified by differential expression analysis and WGCNA, respectively; column G: BG-specific genes identified by RNA-seq and inspection of Allen Brain Atlas. (C) BG-specific genes identified by intersecting of DE genes (third spreadsheet). Column A and B: gene symbol of mouse and human genes; column C: entrez gene ID; column D: ensembl gene ID; column D: description of gene name; column F-H: log2 fold changes. (D) BG-specific genes identified by WGCNA (fourth spreadsheet). Column A and B: gene symbol of mouse and human genes; column C: description of gene name; column D: initial assignment of co-expression module; column E: module membership value in the BG (red) module; column K and L: bRG-specific genes identified by Pollen et al and SINCERA (see *Figure 1A*); column M: BG-specific genes identified by differential gene expression analysis; column N: specific expression in BG of mouse cerebella at postnatal day 56 according Allen Brain Atlas. Scoring criteria: 0, no expression; 1, expressed in Purkinje cells; 2, expression in Purkinje cell layer but not sure in which cell type; 3, expressed in Bergmann glia.

• Supplementary file 2. RNA-seq analysis of microdissected cerebellar tissues of wild-type and *Ptpn11*-cKO embryos with or without *Mek1*[DD] expression at E12.5, E13.5 and E14.5. (A) Summary of RNA-seq (first spreadsheet). (B) Output of STAR mapping (second spreadsheet). (C) Raw gene counts (third spreadsheet). (D) Differentially expressed genes identified by DESeq2 (third spreadsheet). Column A: ensembl gene ID B: gene symbol; column C: entrez gene ID; column D: gene symbol of the human orthologs; column E: entrez ID of human orthologs; Column F: description of gene name; Column G: the mean of normalized counts of all samples; Column H: fold changes on a logarithmic scale to base 2; columns I: Wald statistic; column K: Wald test p-value; column L: Benjamini-Hochberg adjusted p-value; column M: key of pairwise comparison.

• Supplementary file 3. Identification of early-response genes induced by ERK activation. (A) List of differentially expressed genes. Column A: gene symbol; Column B: human gene symbol; column C: ensembl gene ID; column D: description of gene name; Column E: fold changes on a logarithmic scale to base 2; Column F: the mean of normalized expression of all samples; Column G: moderated t-statistic; Column H: raw p-value; Column I: adjusted p-value; column J: log odds that the gene is differentially expressed; column K: key of pairwise comparison.

• Supplementary file 4. Output of gene set enrichment analysis of gene expression in neocortical aRG between humans and mice. (A) Lists of gene sets that are significantly enriched in human aRG (first spreadsheet) or mouse aRG (second spreadsheet). (B) Output of pathway preservation in human and mouse cortical coexpression networks (third spreadsheet).

• Supplementary file 5. List of antibodies, primers used in the current study. (A) Lists of primary and secondary antibodies used in the current study (first spreadsheet). (B, C) Lists of primers used for RT-qPCR (second spreadsheet) and in situ hybridization (third spreadsheet). Gene name, primer names, primer sequence, and references to previously published work (if any) are listed in the first, second, third and fourth column, respectively.

## Major datasets

The following datasets were generated:

| Author(s) | Year | Dataset title | Dataset URL | Database, license, and accessibility information |
|---|---|---|---|---|
| James YH Li | 2016 | RNA-seq Analysis of Wild Type and Ptpn11-deficient cerebellar Transcriptomes | https://www.ncbi.nlm.nih.gov/geo/query/acc.cgi?acc=GSE87104 | Publicly available at the NCBI Gene Expression Omnibus (accession no: GSE87104). |
| Ayoub AE, Oh S, Xie Y, Leng J, Cotney J, Dominguez MH, Noonan JP, Rakic P | 2011 | Transcriptional programs in transient embryonic zones of the cerebral cortex defined by high-resolution mRNA-sequencing | https://www.ncbi.nlm.nih.gov/geo/query/acc.cgi?acc=GSE30765 | Publicly available at the NCBI Gene Expression Omnibus (accession no: GSE30765) |
| Fietz SA, Huttner WB, Pääbo S | 2012 | Transcriptomes of germinal zones of human and mouse fetal neocortex suggest a role of extracellular matrix in progenitor self-renewal | https://www.ncbi.nlm.nih.gov/geo/query/acc.cgi?acc=GSE38805 | Publicly available at the NCBI Gene Expression Omnibus (accession no: GSE38805). |
| Florio M, Albert M, Huttner WB | 2015 | Human-specific gene ARHGAP11B promotes basal progenitor amplification and neocortex expansion | https://www.ncbi.nlm.nih.gov/geo/query/acc.cgi?acc=GSE65000 | Publicly available at the NCBI Gene Expression Omnibus (accession no: GSE65000). |
| Walsh CA, Johnson MB, Wang PP | 2015 | Single Cell Analysis Reveals Unexpected Transcriptional Heterogeneity of Neural Progenitors in the Developing Human Cortex | https://www.ncbi.nlm.nih.gov/geo/query/acc.cgi?acc=GSE66217 | Publicly available at the NCBI Gene Expression Omnibus (accession no: GSE66217). |
| Levi BP, Mich JK, | 2015 | Fixed single-cell transcriptomic | https://www.ncbi.nlm. | Publicly available at |

| Yao Z, Thomsen ER, Hodge RD, Doyle AM, Jang S, Shehata S, Nelson AM, Shapovalova NV, Ramanathan S | | characterization of human radial glial diversity | nih.gov/geo/query/acc. cgi?acc=GSE71858 | the NCBI Gene Expression Omnibus (accession no: GSE71858). |
|---|---|---|---|---|
| Hamilton WB, Brickman J | 2015 | Expression data from mouse embryonic stem cells across a time course of Erk stimulation | https://www.ncbi.nlm. nih.gov/geo/query/acc. cgi?acc=GSE59755 | Publicly available at the NCBI Gene Expression Omnibus (accession no: GSE59755). |

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
