## [Decision Letter]

Thank you for submitting your article "Analogous mechanism regulating formation of neocortical basal radial glia and cerebellar Bergmann glia" for consideration by *eLife*. Your article has been reviewed by three peer reviewers, and the evaluation has been overseen by a Reviewing Editor and Robb Krumlauf as the Senior Editor. The reviewers have opted to remain anonymous.

The reviewers have discussed the reviews with one another and the Reviewing Editor has drafted this decision to help you prepare a revised submission.

The study from Victor Li's group has been thoroughly reviewed by three experts in the field. The study is appreciated for both the comprehensive analysis of gene expression and the analysis of the function of the ERK pathway in genesis of bRG cells. However, there are major concerns regarding the validity of some of the markers, due to method of presentation of the validation data and/or the way in which the validation experiments were done. Since it is the use of these markers that is fundamental to much of the interpretations and the major conclusions, the authors need to address the major concerns regarding the markers. They must also address the other major criticisms raised by the reviewers, several of which cite common concerns. In addition to validation of the markers, they need to show better images of electroporated cells so that one can interpret their morphology and identity, to cite the appropriate ages for each RNA sample and age for validation, to justify use of markers that are expressed more broadly than e.g. only in bRG, etc. In addition, the authors should address the issues raised regarding proper citations throughout the manuscript, and comment more substantively regarding previous and conflicting studies, e.g. the last two major comments by reviewer #3. Many comments can be addressed by rewriting parts of the manuscript, but others might require additional experiments, depending upon whether e.g. tissue can be re-imaged and/or restained for better presentation of marker validation.

*Reviewer #1:*

This is a very interesting paper that explores the gene expression signatures and signaling pathways shared between Bergmann glia in the cerebellum and basal or outer radial glia in the neocortex. Both cell types are glial-like, and BG have been shown by the authors to be critical for folding of the cerebellum, and bRG have been suggested to regulate gyrification in the primate cortex. Thus, although BG do not produce neurons, the comparison is highly relevant. The authors indeed find that many bRG-specific genes also appear to be expressed in BG, and that ERK signaling regulates bRG, as they had previously shown for BG. The results are highly relevant the data in general is of high quality.

I nevertheless have several concerns that need to be addressed, mainly related to interpretation of the data and lack of clarity. One general point is that the authors use the term BG very loosely to mean what is likely a population of several cell types, some being dividing precursors and some being mature postmitotic BG, and each might be distinct at different stages. The second is that throughout the paper the authors do not clarify what age the gene expression analysis is from, and discuss whether it is relevant to mix and match early embryonic and postnatal stages. Finally, it is not clear how the authors assign their number system for expression in the BG layer, in what they think are BG cells alone, or BG and Purkinje cells. Double labeling for critical markers would be very valuable. Also, the authors should determine whether the cells in the BG layer that express the markers are dividing precursors, and whether the markers are expressed at all stages.

*Reviewer #2:*

Heng and colleagues use previously published RNA-Seq data and novel bioinformatics to describe the gene expression typifying neocortical basal radial glia (bRG) and cerebellar Bergmann glia (BG). This study shows quite clearly that a set of genes highly enriched in nascent BG are also present in bRG and furthermore that their apical RG and EGC precursors likely share gene expression profiles. In addition, through comparative studies between mouse and human datasets, the authors demonstrate that ERK signaling is particularly enriched in aRG during their transition to bRG and BG in both mouse and human, but that ERK expression is much more highly expressed in human development. Their studies where forced higher ERK signaling in mouse aRG lead to higher production of bRG clearly support the biological significance of their findings. The study reaches some important conclusions alongside this important gene expression profiling, including that increased bRG are not sufficient to induce folding or gyrification in the mouse neocortex, as has previously been suggested. Overall, the manuscript is very well written, save for a few typographical errors that I am sure the editorial staff can identify. In fact, this is the best-crafted study I have reviewed all year, and I mean to include 2016 with that as well.

*Reviewer #3:*

This paper by Heng and colleagues identifies molecular commonalities between bRG cells in the embryonic cerebral cortex, important for cortical folding, and Bergman Glia cells in the developing cerebellum, important for cerebellar folding. They identify the ERK pathway as potentially important to regulate the formation of these two cell types, and then perform functional experiments and additional transcriptomic meta-analyses to test their hypothesis. They conclude that activation of the FGF-ERK-Etv pathway is important for the formation of bRG and BG cells, and hence for the expansion and folding of these two tissues. The hypothesis is fascinating and the topic timely, the experiments well planned and executed, and the results nicely explained. Unfortunately, there are a number of serious concerns that lessen my initial enthusiasm for this study, including some of their key interpretations. Following is my detailed list of specific concerns and criticisms:

Introduction:

There are serious limitations in the background knowledge of this topic by the authors, which seem to ignore some of the seminal and landmark studies and discoveries.

Introduction, first paragraph – Splitting of SVZ into ISVZ and OSVZ has never been demonstrated to result from increased proliferation of IPCs – this sentence should be eliminated.

Introduction, first paragraph – The concept that OSVZ may be responsible for the tangential expansion of neocortical surface area was first proposed by the original article Reillo et al. 2011 (Cerebral Cortex – published online before the review Lui et al. 2011), and then extensively elaborated in Borrell and Reillo 2012 (Cerebral Cortex). These papers should be cited alongside with Lui et al. 2011 at this point.

Introduction, second paragraph – A recent paper should be cited here, which is explicitly dedicated to the mechanisms of generation of bRG from aRGs: Martínez-Martínez et al. 2016 (Nat Comm).

Introduction, second paragraph – The following papers should be added as reference to the hypothesis that bRG expansion is responsible for cortical folding, as they were seminal for this idea: Reillo et al. 2011 (Cerebral Cortex), Borrell and Reillo 2012 (Cerebral Cortex), Nonaka-Kinoshita et al. 2013 (EMBO J). These ideas were carefully reviewed and updated in Borrell and Götz 2014 (Cur Opin Neurobiol), Taverna et al. 2015 (Annu Rev Neurobiol) and Fernández et al. 2016 (EMBO J).

“However, the contribution of bRG to cortical gyrification remains unclear because bRG abundance does not correlate with either gyrification or the phylogeny of the neocortex (García-Moreno et al., 2012; Hevner and Haydar, 2012; Kelava et al., 2012)” – This statement is absolutely incorrect. Whereas early studies, such as those cited here, showed that the mere presence of bRG in the cerebral cortex is not sufficient to drive cortical folding, several other studies by multiple labs have demonstrated that the relative abundance of bRGs DOES correlate with the degree of cortical folding, and that the phylogeny of cortical folding is much more complex than here implied: Reillo and Borrell 2012 (Cerebral Cortex), Pilz et al. 2013 (Nat Comm), Kelava et al. 2012 (Cerebral Cortex), Kelava et al. 2013 (Front Neuroanat), Lewitus et al. 2014 (PLoS Biol). Remarkably, this sentence is contradicted by the authors themselves two sentences later.

Introduction, second paragraph – The authors should cite the following papers as knowledge on cellular and molecular mechanisms that control the transition of aRG into bRG, published before their study: LaMonica et al. 2013 (Nat Comm), Martinez-Martinez et al. 2016 (EMBO J).

Results:

Gene expression analysis in BG cells – According to the Methods section, transcriptomic analyses were performed from bulk tissue (microdissected layers from thick tissue slices), not single cells, so the analysis contains a very noisy mixture of BG and Purkinje cells. For validation by ISH at postnatal stages, the authors indicate to have distinguished BG from Purkinje cells by morphology, allowing them to then confirm the gene expression analysis. However, there is not a single example of this. Only low power panoramic images are provided, from where it is impossible to distinguish BG from Purkinje cells. This is true throughout the manuscript when the authors assess gene expression patterns using Allen Brain Atlas, where images do not provide cell type-specific information.

In this regard, Figure 1 shows images for 1 gene only (!), at embryonic stage instead of postnatal as shown in Figure 1, and where the control image (*Ptpn11*-cKO) is from a different embryonic stage, which clearly is not appropriate for comparison. Moreover, for a number of genes, expression is not specific to BG but clearly extended to the granule cell layer, such as Atp1a2, Cst1, Tubb2a, Tubb2b, *Fgfr1*.

In the validation by analysis of KO embryos, Figure 2—figure supplement 1, panel B shows clearly that KO embryos are completely missing ISH signal found in wt embryos. However, expression for these genes is also lost from all the surrounding tissues, not only the cerebellar anlage. How can this be interpreted as these genes being specific to BG?

Obviously, the clarity and specificity of all these analyses must necessarily be convincing, as these are the foundations for the rest of the study. It would have been much more appropriate to perform single cell RNAseq for this study. However, given that this is now out of the scope of this manuscript, it may also be of help to provide confocal images demonstrating co-expression at single cell level between BG markers and a collection of the newly identified genes (not merely 1 gene). Also, it will help the credibility of this story to add data from Purkinje cells (negative expression) in analyses like, for example, those in Figure 1, Figure 2, Figure 2—figure supplement 1.

In some of the tissue stains, it is worrisome to note that Hopx is detected at high levels in embryonic mouse cortical VZ (Figure 5—figure supplement 1, Figure 10). Does this not seriously question the reliability of Hopx as a gold-standard bRG marker used in the transcriptomic analyses of this study? The same argument is also true for the identification of bRG across this study (i.e. in Figure 5 and Figure 10). This has very profound consequences in the interpretation of results all along the manuscript, as well as its Discussion.

In the analysis of ERK activity, the authors first show that phosphorylated ERK is found only in the rostral-most aspect of the mouse cortex (Figure 3), but then in Figure 4 they show that expression of Clu (the only ERK-early response gene expressed in mouse cortex at the same embryonic age) is lowest in the rostral aspect, graded to highest in the caudal region. How is this contradiction explained?

Induction of bRG formation by ERK activation – Images to illustrate the Mek electroporation experiments (not the controls!) are always shown from the cortical plate (without including the IZ or SVZ), which is an abnormal layer to find bRG cells. This leads to the parsimonious conclusion that these GFP+ cells in Mek-electroporated embryos are completely abnormal and ectopic, possibly with little resemblance to bRGs. Moreover, Figure 5 are claimed to illustrate the presence of abundant bRGs, including their long basal process; however, the cells shown are (again) quite close to the pial surface, most likely already within the cortical plate, so they may well be just migrating neurons reaching the pial surface and that, by virtue of MEK overactivation, continue expressing markers of RG cells.

Overexpression of constitutively-active Mek1 by in utero electroporation has been performed previously, even at the same embryonic stages as in this study, with some very different results (Li et al. 2012, Neuron 75:1035-1050): no horizontal clusters of GFP^+^ cells, and abundant GFP^+^ astrocytes in all cortical layers. The authors need to reconcile their findings with these results published previously. In Discussion the authors make some comments regarding this paper, but not related to this question.

Overexpression of Mek1DD does not lead to cortical folding in mouse, so the authors conclude that bRG expansion itself is not sufficient to induce folding of the mouse cortex. However, it has been previously demonstrated, by several labs independently, that bRG amplification DOES lead to cortical folding in mouse (Stahl et al. 2013, Florio et al. 2015), and increased folding in ferret (Nonaka-kinoshita et al. 2013). Given my above significant concerns about the bRG identity of cells overproduced upon Mek overexpression, is it not possible that this conclusion is wrong? Definitely this reviewer is not convinced that electroporation of Mek1DD causes the overproduction of morphologically-distinguishable bRGs (though proliferative, Pax6^+^ cells are indeed present). Moreover, the authors have not checked for cell death, which may eliminate part of the overproduced bRGs and hence prevent subsequent cortical folding.

There are a lot of results presented as "Data not shown". I don't understand the need to not show many of results, given that there are almost no space limitations in *eLife*, including supplementary figures.

Figures

Figure 1 – Images in panels C and D are of insufficient resolution, not acceptable.

Figure 4 – There are two panels labeled as "C".

Figure 5 – The resolution of images in insufficient to distinguish basal and apical processes in the GFP^+^ cells.

Figure 6 – The low resolution of most of images presented is unacceptable.

Figure 9 -Fabp7 protein is nearly absent in cKO embryos, so its absence from electroporated cells means bears no real additional significance. Remarkably Fabp7 expression is rescued in these mutants upon sparse electroporation of Mek1DD or Etv4, with a very significant cell non-autonomous effect. How is this explained? Is it a question of photographic exposure during image acquisition?

Figure 9 – Panels for GFP, Fabp7 and Sox9 labeling show the exact same image, which must be replaced by the real experiment images.

---

## [Author Response]

*Reviewer #3:*

*This paper by Heng and colleagues identifies molecular commonalities between bRG cells in the embryonic cerebral cortex, important for cortical folding, and Bergman Glia cells in the developing cerebellum, important for cerebellar folding. They identify the ERK pathway as potentially important to regulate the formation of these two cell types, and then perform functional experiments and additional transcriptomic meta-analyses to test their hypothesis. They conclude that activation of the FGF-ERK-Etv pathway is important for the formation of bRG and BG cells, and hence for the expansion and folding of these two tissues. The hypothesis is fascinating and the topic timely, the experiments well planned and executed, and the results nicely explained. Unfortunately, there are a number of serious concerns that lessen my initial enthusiasm for this study, including some of their key interpretations. Following is my detailed list of specific concerns and criticisms:*

*Introduction:*

*There are serious limitations in the background knowledge of this topic by the authors, which seem to ignore some of the seminal and landmark studies and discoveries.*

*Introduction, first paragraph – Splitting of SVZ into ISVZ and OSVZ has never been demonstrated to result from increased proliferation of IPCs – this sentence should be eliminated.*

Changes are made.

*Introduction, first paragraph – The concept that OSVZ may be responsible for the tangential expansion of neocortical surface area was first proposed by the original article Reillo et al. 2011 (Cerebral Cortex – published online before the review Lui et al. 2011), and then extensively elaborated in Borrell and Reillo 2012 (Cerebral Cortex). These papers should be cited alongside with Lui et al. 2011 at this point.*

These references are added.

*Introduction, second paragraph – A recent paper should be cited here, which is explicitly dedicated to the mechanisms of generation of bRG from aRGs: Martínez-Martínez et al. 2016 (Nat Comm).*

This reference is added.

*Introduction, second paragraph – The following papers should be added as reference to the hypothesis that bRG expansion is responsible for cortical folding, as they were seminal for this idea: Reillo et al. 2011 (Cerebral Cortex), Borrell and Reillo 2012 (Cerebral Cortex), Nonaka-Kinoshita et al. 2013 (EMBO J). These ideas were carefully reviewed and updated in Borrell and Götz 2014 (Cur Opin Neurobiol), Taverna et al. 2015 (Annu Rev Neurobiol) and Fernández et al. 2016 (EMBO J).*

The references are added.

*“However, the contribution of bRG to cortical gyrification remains unclear because bRG abundance does not correlate with either gyrification or the phylogeny of the neocortex (García-Moreno et al., 2012; Hevner and Haydar, 2012; Kelava et al., 2012)” – This statement is absolutely incorrect. Whereas early studies, such as those cited, here showed that the mere presence of bRG in the cerebral cortex is not sufficient to drive cortical folding, several other studies by multiple labs have demonstrated that the relative abundance of bRGs DOES correlate with the degree of cortical folding, and that the phylogeny of cortical folding is much more complex than here implied: Reillo and Borrell 2012 (Cerebral Cortex), Pilz et al. 2013 (Nat Comm), Kelava et al. 2012 (Cerebral Cortex), Kelava et al. 2013 (Front Neuroanat), Lewitus et al. 2014 (PLoS Biol). Remarkably, this sentence is contradicted by the authors themselves two sentences later.*

In the previous experiments, the induction of bRG is associated with expansion of the IPC. Therefore, these experiments do not definitely tell us that increased generation of bRG alone is responsible for cortical folding.

*Introduction, second paragraph – The authors should cite the following papers as knowledge on cellular and molecular mechanisms that control the transition of aRG into bRG, published before their study: LaMonica et al. 2013 (Nat Comm), Martinez-Martinez et al. 2016 (EMBO J).*

These references are added.

*Results:*

*Gene expression analysis in BG cells – According to the Methods section, transcriptomic analyses were performed from bulk tissue (microdissected layers from thick tissue slices), not single cells, so the analysis contains a very noisy mixture of BG and Purkinje cells. For validation by ISH at postnatal stages, the authors indicate to have distinguished BG from Purkinje cells by morphology, allowing them to then confirm the gene expression analysis. However, there is not a single example of this. Only low power panoramic images are provided, from where it is impossible to distinguish BG from Purkinje cells. This is true throughout the manuscript when the authors assess gene expression patterns using Allen Brain Atlas, where images do not provide cell type-specific information.*

We add a new figure to demonstrate that the loss of Ptpn11 blocks BG formation but spare the other major cerebellar cell types (Figure 2—figure supplement 2). Although we cannot rule out that transcription of some genes in non-BG lineages could be affected by Ptpn11 deletion, we expect that BG-specific genes will be greatly reduced in *Ptpn11*-cKO cerebella. We examined additional BG markers and showed that the transcripts of the genes were greatly reduced in *Ptpn11*-cKO cerebella (Figure 1—figure supplement 1).

We have modified the figures with the highest available resolution of ISH images from the Allen Brain Atlas (Figure 1, Figure 1—figure supplement 1 and Figure 1—figure supplement 2).

*In this regard, Figure 1 shows images for 1 gene only (!), at embryonic stage instead of postnatal as shown in Figure 1, and where the control image (Ptpn11-cKO) is from a different embryonic stage, which clearly is not appropriate for comparison. Moreover, for a number of genes, expression is not specific to BG but clearly extended to the granule cell layer, such as Atp1a2, Cst1, Tubb2a, Tubb2b, Fgfr1.*

We have added additional stages to Figure 1 and additional markers analysis in a new figure (Figure 1—figure supplement 1). We have made changes to clarify that the identified BG candidate markers are highly expressed (but not exclusively) or specifically expressed in BG (Figure 1 legend).

*In the validation by analysis of KO embryos, Figure 2—figure supplement 1, panel B shows clearly that KO embryos are completely missing ISH signal found in wt embryos. However, expression for these genes is also lost from all the surrounding tissues, not only the cerebellar anlage. How can this be interpreted as these genes being specific to BG?*

We have added a section to discuss the potential pitfall of our bioinformatic approach. As expected, our strategy will likely identify downstream targets of the Ptpn11-ERK pathway that are essential for BG formation (possibly other unknown aspects of cerebellar development), and some of them are not necessarily expressed or maintained in the developing BG. However, we demonstrate that the resulted gene lists are significantly enriched for genes that are expressed in the BG lineage.

*Obviously, the clarity and specificity of all these analyses must necessarily be convincing, as these are the foundations for the rest of the study. It would have been much more appropriate to perform single cell RNAseq for this study. However, given that this is now out of the scope of this manuscript, it may also be of help to provide confocal images demonstrating co-expression at single cell level between BG markers and a collection of the newly identified genes (not merely 1 gene). Also, it will help the credibility of this story to add data from Purkinje cells (negative expression) in analyses like, for example, those in Figure 1,Figure 2, Figure 2—figure supplement 1.*

We have now examined additional genes (Figure 1—figure supplement 1). Furthermore, we add confocal images to show that Hopx is completely colocalized with Fabp7 and Sox2 in E14.5 and E18.5 cerebella (Figure 1). As Fabp7 and Sox2 are not expressed in Purkinje cells, we don’t believe that is necessary to analyze Hopx and Purkinje cell markers.

*In some of the tissue stains, it is worrisome to note that Hopx is detected at high levels in embryonic mouse cortical VZ (Figure 5—figure supplement 1, Figure 10). Does this not seriously question the reliability of Hopx as a gold-standard bRG marker used in the transcriptomic analyses of this study? The same argument is also true for the identification of bRG across this study (i.e. in Figure 5 and Figure 10). This has very profound consequences in the interpretation of results all along the manuscript, as well as its Discussion.*

Identification of bRG could be ambiguous in some cases due to the limited number of specific markers. Our study is, to our knowledge, the first to take full advantage of the Pollen et al. and Thomsen et al. single-cell RNA-seq studies to characterize the molecular features of the induced bRG, and therefore will provide a richer resource for future studies regarding bRG. Regarding the specificity of Hopx, we have revised the text to indicate that Hopx is “a specific” instead of “the most specific” bRG marker (subsection “bRG-specific genes are enriched in BG”, first paragraph). We also emphasize that our analyses were based on a battery of known markers, self-renewal potential, and morphology to ascertain the induction of bRG in overexpression experiments (Figure 5 and Figure 10).

*In the analysis of ERK activity, the authors first show that phosphorylated ERK is found only in the rostral-most aspect of the mouse cortex (Figure 3), but then in Figure 4 they show that expression of Clu (the only ERK-early response gene expressed in mouse cortex at the same embryonic age) is lowest in the rostral aspect, graded to highest in the caudal region. How is this contradiction explained?*

*Clu* is only one of the ERK-early response genes according the microarray study, but other signaling pathways can also regulate its expression. The presence of *Clu* in the caudal region does not necessarily indicate the presence of strong ERK activity in this region. Therefore, we don’t believe that there is a contradiction here.

*Induction of bRG formation by ERK activation – Images to illustrate the Mek electroporation experiments (not the controls!) are always shown from the cortical plate (without including the IZ or SVZ), which is an abnormal layer to find bRG cells. This leads to the parsimonious conclusion that these GFP+ cells in Mek-electroporated embryos are completely abnormal and ectopic, possibly with little resemblance to bRGs. Moreover, Figure 5 are claimed to illustrate the presence of abundant bRGs, including their long basal process; however, the cells shown are (again) quite close to the pial surface, most likely already within the cortical plate, so they may well be just migrating neurons reaching the pial surface and that, by virtue of MEK overactivation, continue expressing markers of RG cells.*

We have now added images to show the induced bRG in the lower layer (Figure 5—figure supplement 1). Unlike primates, mice do not have the outer SVZ in the developing neocortex. It is thus difficult to predict to where the induced bRG should be confined. In the control embryos, we do find bRG-like cells that are positive for pERK, Hopx, Sox2, and Fabp7 in the cortical plates (Figure 4). In the Discussion, we add a section to discuss the implication of the distribution of Mek1DD-induced bRG in the mouse cortex (Discussion, sixth paragraph).

*Overexpression of constitutively-active Mek1 by in utero electroporation has been performed previously, even at the same embryonic stages as in this study, with some very different results (Li et al. 2012, Neuron 75:1035-1050): no horizontal clusters of GFP^+^ cells, and abundant GFP^+^ astrocytes in all cortical layers. The authors need to reconcile their findings with these results published previously. In Discussion the authors make some comments regarding this paper, but not related to this question.*

We have made changes to compare these two studies (Discussion, fourth paragraph). The previous study did not examine any bRG markers except for Fabp7, and the histological data were from P22 and P60, much later than what we did in the current study. Therefore, we cannot comment on how different our results really are. However, different experimental procedure including the timing of in utero electroporation and different expression vectors could certainly contribute to any potential differences in these two studies.

*Overexpression of Mek1DD does not lead to cortical folding in mouse, so the authors conclude that bRG expansion itself is not sufficient to induce folding of the mouse cortex. However, it has been previously demonstrated, by several labs independently, that bRG amplification DOES lead to cortical folding in mouse (Stahl et al. 2013, Florio et al. 2015), and increased folding in ferret (Nonaka-kinoshita et al. 2013). Given my above significant concerns about the bRG identity of cells overproduced upon Mek overexpression, is it not possible that this conclusion is wrong? Definitely this reviewer is not convinced that electroporation of Mek1DD causes the overproduction of morphologically-distinguishable bRGs (though proliferative, Pax6^+^ cells are indeed present). Moreover, the authors have not checked for cell death, which may eliminate part of the overproduced bRGs and hence prevent subsequent cortical folding.*

We now emphasize that the manipulation in previous studies causes overproduction of both bRG and IPC, whereas activation of the FGF-ERK-ETV axis induces bRG but not IPC.

*There are a lot of results presented as "Data not shown". I don't understand the need to not show many of results, given that there are almost no space limitations in eLife, including supplementary figures.*

We have now added all the data that are essential for our conclusion (see above).

*Figures*

*Figure 1 – Images in panels C and D are of insufficient resolution, not acceptable.*

These images are replaced by high-resolution confocal images.

*Figure 4 – There are two panels labeled as "C".*

Correction is made.

*Figure 5 – The resolution of images in insufficient to distinguish basal and apical processes in the GFP^+^ cells.*

For the initial submission, the quality of the many images was regrettably compromised. We now ensure high quality images are used for the resubmission. In addition, we have now included a three-dimensional reconstruction of induced bRG (Video 1).

*Figure 6 – The low resolution of most of images presented is unacceptable.*

Unfortunately, the confocal microscope of our facility lacks the laser to detect one of the fluorescent probes. We have to use a wide-field microscope for some of the images in this figure. To improve the clarity, we have enlarged the images and make sure that the image quality is maintained in the submission process.

*Figure 9 -Fabp7 protein is nearly absent in cKO embryos, so its absence from electroporated cells means bears no real additional significance. Remarkably Fabp7 expression is rescued in these mutants upon sparse electroporation of Mek1DD or Etv4, with a very significant cell non-autonomous effect. How is this explained? Is it a question of photographic exposure during image acquisition?*

We have presented a negative control to show the background level of Fabp7 (BLBP) expression in both control and mutant cerebella (Figure 9). As we described in the previous publication (Li et al., 2014), Fabp7 is sporadically expressed in cells in the VZ or immediately above the VZ of *Ptpn11*-cKO cerebella after E14.5 (also see Figure 1—figure supplement 1). However, these Fabp7^+^ cells lack the characteristic BG morphology in the mutants as they extend their cellular processes parallel, instead of perpendicular, to the VZ. Here, we show that the rescued BG identified by molecular markers and morphology were mostly derived from GFP^+^ cells, suggesting that Mek1DD, Etv4 and Etv5 likely act cell-autonomously.

*Figure 9 – Panels for GFP, Fabp7 and Sox9 labeling show the exact same image, which must be replaced by the real experiment images.*

We have corrected the mistake.